# Towards Stable, Globally Expressive Graph Representations with Laplacian Eigenvectors

## Abstract

Graph neural networks (GNNs) have achieved remarkable success in a variety of machine learning tasks over graph data. Existing GNNs usually rely on message passing, i.e., computing node representations by gathering information from the neighborhood, to build their underlying computational graphs. Such an approach has been shown fairly limited in expressive power, and often fails to capture global characteristics of graphs. To overcome the issue, a popular solution is to use Laplacian eigenvectors as additional node features, as they are known to contain global positional information of nodes, and can serve as extra node identifiers aiding GNNs to separate structurally similar nodes. Since eigenvectors naturally come with symmetries—namely, $O(p)$-group symmetry for every $p$ eigenvectors with equal eigenvalue, properly handling such symmetries is crucial for the stability and generalizability of Laplacian eigenvector augmented GNNs. However, using a naive $O(p)$-group invariant encoder for each $p$-dimensional eigenspace may not keep the full expressivity in the Laplacian eigenvectors. Moreover, computing such invariants inevitably entails a hard split of Laplacian eigenvalues according to their numerical identity, which suffers from great instability when the graph structure has small perturbations. In this paper, we propose a novel method exploiting Laplacian eigenvectors to generate *stable* and globally *expressive* graph representations. The main difference from previous works is that (i) our method utilizes **learnable** $O(p)$-invariant representations for each Laplacian eigenspace of dimension $p$, which are built upon powerful orthogonal group equivariant neural network layers already well studied in the literature, and that (ii) our method deals with numerically close eigenvalues in a **smooth** fashion, ensuring its better robustness against perturbations. Experiments on various graph learning benchmarks witness the competitive performance of our method, especially its great potential to learn global properties of graphs.

## 1 Introduction

Numerous real-world data—such as molecules, electric circuits or social networks—can be represented by graphs. Machine learning over graphs is thus an important approach to find underlying relations among them, and make predictions concerning novel data. So far, graph neural networks (GNNs) have proved successful on a plethora of learning tasks over graphs (Wu et al., 2020; Zhou et al., 2020), spanning across domains such as chemistry (Deshpande et al., 2002; Jin et al., 2018; Reiser et al., 2022), biology (Stokes et al., 2020; Zitnik & Leskovec, 2017; Zitnik et al., 2018), social recommendations (Ying et al., 2018) or electronic design automation (Lopera et al., 2021).

One of the most popularly adopted GNN architecture is message passing neural network (MPNN), which maintains a representation vector $\boldsymbol{h}_u$ for each node $u$, and iteratively updates it by gathering information from the neighboring nodes of $u$. Despite its relative simplicity and efficiency, it has several weaknesses that severely limit its performance. One important problem is its **limited expressive power**, referring to the fact that MPNNs often fail to distinguish between two non-isomorphic graphs, or two structurally different nodes with similar neighborhood configuration (Xu et al., 2018; Zhang et al., 2021). Another issue is its **inability to capture global properties** of graphs, meaning that it cannot truthfully learn long-range interactions within a graph, due to "oversquashing" that occurs as a result of multiple message passing steps (Alon & Yahav, 2020; Dwivedi et al., 2022).

In this paper, we refer to the former problem as a lack of **local** expressive power, while the latter as one concerning **global** expressive power.

A great number of works have attempted to tackle the aforementioned weaknesses of MPNNs, which we will review in Section 5 and Appendix C. For now, we restrict our discussion to one specific approach—using Laplacian eigenvectors as node feature augmentations. Why are we particularly interested in it? The reason is that Laplacian eigenvectors may alleviate **both** issues we raise above. First, Laplacian eigenvectors provably contain rich **local structural information** (Cvetković et al., 1997; Fürer, 2010; Rattan & Seppelt, 2023), and can thus serve as additional node labels, making it easier for a GNN to separate nodes that are otherwise similar. Furthermore, they can reflect the absolute position of each node within the graph (Von Luxburg, 2007), making GNNs aware of potential **long-range interactions**. Indeed, a vast literature has regarded Laplacian eigenvectors as so-called *graph positional encodings*, which play important roles both in MPNNs and graph transformers (Dwivedi et al., 2021; 2023; Rampášek et al., 2022; Ying et al., 2021b).

Although Laplacian eigenvectors provide a promising solution for expressive graph representation learning, there are some well-known constraints that one must take into account for their reliable use. The first is **orthogonal-group invariance**. As is first pointed out by Wang et al. (2022); Lim et al. (2022), given a Laplacian $L$, its eigen-decomposition is in general not unique. In fact, assuming that $v_1, \ldots, v_p$ are $p$ mutually orthogonal normalized eigenvectors of a Laplacian $L$ that correspond to the same eigenvalue $\lambda$, then so are $v'_1, \ldots, v'_p$, as long as the two groups of eigenvectors can be associated via a $p \times p$ orthogonal matrix $Q$, namely $V' = V \cdot Q$ where $V = (v_1, \ldots, v_p)$ and $V' = (v'_1, \ldots, v'_p)$. One must ensure that the network output is invariant to such orthogonal transformations, so as to produce identical representations for identical (i.e., isomorphic) graphs. Another related but stricter constraint is **stability**, which, as formally defined by Huang et al. (2024), demands that network outputs should be close when the input graph undergoes small perturbations. It is easy to see that orthogonal-group invariance is a special case of stability in which the strength of perturbation approaches zero.

To ensure orthogonal-group invariance (and furthermore, stability), Lim et al. (2022) and Huang et al. (2024) both propose to extract spectral information from *inner products* between Laplacian eigenvectors—namely, $VV^T$ with $V = (v_1, \ldots, v_p)$ being the matrix consisting of $p$ mutually orthogonal normalized eigenvectors within an eigenspace of dimension $p$—instead of the eigenvectors $(v_1, \ldots, v_p)$ themselves. The inner product matrices $VV^T$ for different Laplacian eigenspaces are then processed by invariant graph networks (IGNs) proposed in (Maron et al., 2018; 2019) to produce node feature augmentations. Despite being provably invariant to $O(p)$ transformations and even stable (with carefully designed network architectures), their learning architectures based on inner products are not flexible enough, and may lose much of the rich structural and positional information carried by vanilla Laplacian eigenvectors. Earlier than the above two works, Wang et al. (2022) has proposed a special message passing operation in which only the norms of differences between rows of $V$ are used, and proved its stability. However, this method even dismisses important eigenvalue information by treating Laplacian eigenvectors from different eigenspaces uniformly. Given the limitations of existing methods to utilize Laplacian eigenvectors, a natural question is **whether we can recover the information inherent in vanilla Laplacian eigenvectors while ensuring stability**. To make the question even more general, we may ask:

> *What is the representational limit of graph learning methods exploiting Laplacian eigenvectors, given the stability constraint?*

In this paper, we attempt to partially answer the general question posed above. Our main contributions are summarized below.

- We propose vanilla orthogonal group equivariant augmentation (**Vanilla OGE-Aug**), a novel method exploiting Laplacian eigenvectors to produce node feature augmentations. Inspired by the property of orthogonal group invariance of **invariant point cloud networks** (for example, Tensor Field Network (Thomas et al., 2018) and its variant (Finkelshtein et al., 2022)) as well as their great expressive power, we use them to process the Laplacian eigenvectors, enabling the construction of node feature augmentations much more expressive than previous ones that make use of inner products between eigenvectors.

- We theoretically prove that Vanilla OGE-Aug, combined with an MPNN, can lead to **universal representations** of graphs, as long as the invariant point cloud networks we use are powerful enough. Previous works have theoretically guaranteed the expressive power of several specific invariant point cloud networks, which lays the foundations for the practicality of our theoretical result.

- Although Vanilla OGE-Aug can be maximally expressive, it unfortunately lacks stability. We then propose a **smooth** variant of Vanilla OGE-Aug, namely **OGE-Aug**, by trading expressive power for better stability. Our approach is to use a series of "soft" masks to filter Laplacian eigenvectors that belong to different eigenspaces, instead of hard-splitting and separately processing them. We theoretically prove the stability of OGE-Aug, and evaluate its empirical performance on various real-world graph datasets. The results indicate that our method not only shows competitive performance on popular graph benchmarks, but is surprisingly good at learning **global properties of graphs**.

## 2 PRELIMINARIES

We use $\mathcal{G}$ to denote the set of all simple, undirected graphs. For a graph $G \in \mathcal{G}$, its node set and edge set are denoted by $\mathcal{V}(G)$ and $\mathcal{E}(G)$ respectively. Graphs considered in this paper are usually accompanied with node features, defined as a function from $\mathcal{V}(G)$ to $\mathbb{R}^d$.

For a graph with $n$ nodes labeled by $1, \ldots, n$ respectively, its adjacency matrix is defined as $\boldsymbol{A} \in \{0, 1\}^{n \times n}$ in which $A_{ij} = 1$ if and only if nodes $i$ and $j$ are connected; further, if the graph has node features, the node features are represented by a matrix $\boldsymbol{X} \in \mathbb{R}^{n \times d}$ whose $i$-th row corresponds to the feature of node $i$.

Given the adjacency matrix $\boldsymbol{A} \in \{0, 1\}^{n \times n}$ of graph $G$, we define the Laplacian of graph $G$ as $\boldsymbol{L} = \boldsymbol{D} - \boldsymbol{A}$, in which $\boldsymbol{D} = \operatorname{diag}(d(1), \ldots, d(n))$, with $d(i)$ being the degree of node $i$ ($i = 1, \ldots, n$). It is not hard to see that with $G$ being simple and undirected, its Laplacian $\boldsymbol{L}$ is real symmetric, and further positive semi-definite. Therefore, all eigenvalues of $\boldsymbol{L}$ are real non-negative. One may also verify that 0 is always an eigenvalue of $\boldsymbol{L}$ (thus being the smallest eigenvalue of $\boldsymbol{L}$). If $\boldsymbol{L}$ has an eigenvalue $\lambda$ with multiplicity $\mu$, the linear subspace spanned by the $\mu$ mutually orthogonal eigenvectors of $\boldsymbol{L}$ corresponding to $\lambda$ is called an eigenspace of $\boldsymbol{L}$ with dimension $\mu$.

Let $\boldsymbol{A} \in \mathbb{R}^{n \times n}$. $\boldsymbol{A}$ is said to be orthogonal if $\boldsymbol{A}\boldsymbol{A}^T = \boldsymbol{A}^T\boldsymbol{A} = \boldsymbol{I}$, with $\boldsymbol{I}$ being the identity matrix. Given a positive integer $n$, we use $O(n)$ to denote the set of all orthogonal matrices of shape $n \times n$. A 0-1 matrix $\boldsymbol{A} \in \{0, 1\}^{n \times n}$ is said to be a permutation matrix if each of its rows and columns has exactly one 1-element. Let $S_n$ be the set of all permutation matrices of shape $n \times n$. It's easy to see that $S_n \subseteq O(n)$.

To simplify our discussion below, we further introduce the following shorthands:

- Assume $\{\boldsymbol{V}_1, \ldots, \boldsymbol{V}_k\} \subset \mathbb{R}^{n \times p}$ is a set of $n \times p$ matrices, in which $\boldsymbol{V}_j$ can be row-wise decomposed as $\boldsymbol{V}_j = (\boldsymbol{v}_{j1}, \ldots, \boldsymbol{v}_{jn})^T$, each $\boldsymbol{v}_{ji} \in \mathbb{R}^p$, $i = 1, \ldots, n$. Further let $g$ be a set function, namely $g : 2^{\mathbb{R}^p} \to \mathbb{R}$. Then we use $g(\{\boldsymbol{V}_1, \ldots, \boldsymbol{V}_k\}) \in \mathbb{R}^n$ to denote the vector whose $i$-th component equals $g(\{\boldsymbol{v}_{1i}, \ldots, \boldsymbol{v}_{ki}\})$, for $i = 1, \ldots, n$.

- Given $\boldsymbol{V}_1 \in \mathbb{R}^{n \times p_1}, \ldots, \boldsymbol{V}_k \in \mathbb{R}^{n \times p_k}$, let $\operatorname{concat}[\boldsymbol{V}_1, \ldots, \boldsymbol{V}_k] \in \mathbb{R}^{n \times (p_1 + \cdots + p_k)}$ be the concatenation of $\boldsymbol{V}_1, \ldots, \boldsymbol{V}_k$ along the row dimension.

## 3 UNIVERSAL GRAPH REPRESENTATION WITH LAPLACIAN EIGENVECTORS

Despite the great number of works showing the efficacy of using Laplacian eigenvectors in graph learning tasks, few (Fürer, 2010; Rattan & Seppelt, 2023) have studied theoretically their expressiveness upper-bound—namely, **to what extent can the information of a graph be learned, merely from its Laplacian eigenvectors?** This is a weaker version of the general question we pose in Section 1, with the stability constraint removed. In this section, we will show that the answer to this weaker question is rather optimistic: ignoring the stability constraint, Laplacian eigenvectors can actually lead to **universal representations** of graphs. To reach the point, we start by reconsidering the problem of finding universal graph representations from the perspective of Laplacian eigenvalues and

eigenvectors (Proposition 3.2), and then give a concrete construction of such universal representation (Proposition 3.5).

We first present the definition for universal representations of graphs.

**Definition 3.1** (Universal representation). Let $f$ be a function mapping each pair $(G, \boldsymbol{X}_G)$ to a real value $f(G, \boldsymbol{X}_G) \in \mathbb{R}$, where $G \in \mathcal{G}$ is a graph and $\boldsymbol{X}_G \in \mathbb{R}^{|\mathcal{V}(G)| \times d}$ stands for node features accompanied with $G$. Further let $\boldsymbol{A}_G$ be the adjacency matrix of graph $G$. The function $f$ is said to be a **universal representation** if the following condition holds: for any two pairs $(G, \boldsymbol{X}_G)$ and $(H, \boldsymbol{X}_H)$, $f(G, \boldsymbol{X}_G) = f(H, \boldsymbol{X}_H)$ if and only if $\exists \boldsymbol{P} \in S_{|\mathcal{V}(G)|}$,

$$\boldsymbol{A}_G = \boldsymbol{P} \boldsymbol{A}_H \boldsymbol{P}^T, \quad \boldsymbol{X}_G = \boldsymbol{P} \boldsymbol{X}_H. \tag{1}$$

In other words, $f$ should produce equal outputs only for graphs that are identical up to a permutation of nodes.

Next, we will associate the concept of universal representations with eigendecompositions of graph Laplacians. We denote $\boldsymbol{L}_G$ the Laplacian of a simple, undirected graph $G$. Due to the properties of graph Laplacians (stated in Section 2), we may assume that $\boldsymbol{L}_G$ has $K$ distinct real eigenvalues $\lambda_1, \ldots, \lambda_K$, with $0 = \lambda_1 < \lambda_2 < \cdots < \lambda_K$. We further use $\mu_j$ to denote the multiplicity of eigenvalue $\lambda_j$, and $\boldsymbol{V}_j \in \mathbb{R}^{|\mathcal{V}(G)| \times \mu_j}$ the set of mutually orthogonal normalized eigenvectors corresponding to $\lambda_j$ (each column of $\boldsymbol{V}_j$ being an eigenvector that has $L^2$-norm scaled to 1), for $j = 1, \ldots, K$. Following Fürer (2010), we also denote

$$\operatorname{Spec} G = ((\lambda_1, \mu_1), (\lambda_2, \mu_2), \ldots, (\lambda_K, \mu_K)) \tag{2}$$

the *spectrum* of $G$.

Given the above notations, the following proposition is straightforward.

**Proposition 3.2.** *Let $G, H \in \mathcal{G}$ with $|\mathcal{V}(G)| = |\mathcal{V}(H)|$. Let $\boldsymbol{A}_G$ and $\boldsymbol{A}_H$ be their adjacency matrices respectively. The following two statements are equivalent:*
*(i) $\exists \boldsymbol{P} \in S_{|\mathcal{V}(G)|}, \boldsymbol{A}_G = \boldsymbol{P} \boldsymbol{A}_H \boldsymbol{P}^T$.*
*(ii) Both of the following conditions hold.*

- $\operatorname{Spec} G = \operatorname{Spec} H$.

- *Let the spectrum of $G$ (and thus $H$) be $((\lambda_1, \mu_1), \ldots, (\lambda_K, \mu_K))$, and $\boldsymbol{V}_j, \boldsymbol{V}_j' \in \mathbb{R}^{|\mathcal{V}(G)| \times \mu_j}$ be sets of mutually orthogonal normalized eigenvectors belonging to $G, H$ respectively, both corresponding to eigenvalue $\lambda_j$, for $j = 1, \ldots, K$. There exists $\boldsymbol{P} \in S_{|\mathcal{V}(G)|}$ and $\boldsymbol{Q}_j \in O(\mu_j)$ ($j = 1, \ldots, K$), such that*

$$\boldsymbol{V}_j = \boldsymbol{P} \boldsymbol{V}_j' \boldsymbol{Q}_j. \tag{3}$$

We include the proof in Appendix A. Proposition 3.2 implies that in order to find universal representations of a graph, it may be helpful to find a sufficiently expressive representation for each of its Laplacian eigenspace. Nevertheless, such representation must stay invariant under actions of $O(p)$-group elements for an eigenspace of dimension $p$, due to the existence of arbitrary $\boldsymbol{Q}_j$ matrices ($j = 1, \ldots, K$). Thus, we are motivated to define as following an $O(p)$-invariant universal representation.

**Definition 3.3** ($O(p)$-invariant universal representation). Let $f : \bigcup_{n=0}^{\infty} \mathbb{R}^{n \times p} \to \bigcup_{n=0}^{\infty} \mathbb{R}^{n \times 1}$. Given an input $\boldsymbol{V} \in \mathbb{R}^{n \times p}$, $f$ outputs a column vector $f(\boldsymbol{V}) \in \mathbb{R}^{n \times 1}$. The function $f$ is said to be an $\boldsymbol{O(p)}$**-invariant universal representation** if given $\boldsymbol{V}, \boldsymbol{V}' \in \mathbb{R}^{n \times p}$ and $\boldsymbol{P} \in S_n$, the following two conditions are equivalent: (i) $f(\boldsymbol{V}) = \boldsymbol{P} f(\boldsymbol{V}')$; (ii) $\exists \boldsymbol{Q} \in O(p)$, such that $\boldsymbol{V} = \boldsymbol{P} \boldsymbol{V}' \boldsymbol{Q}$.

By Definition 3.3, an $O(p)$-invariant universal representation is one that assigns an output to each point of a point set embedded in $\mathbb{R}^p$, in a way that is invariant to global $O(p)$ rotations, equivariant to point permutations, and injective with respect to all possible point set configurations. Such networks have been named *universal point cloud networks*, whose design has been intensively studied, as we will survey in Section 5.

We still need another definition which follows Zaheer et al. (2017).

**Definition 3.4** (Universal set representation). Let $\mathcal{X}$ be a non-empty set. A function $f : 2^{\mathcal{X}} \to \mathbb{R}$ is said to be a **universal set representation** if $\forall X_1, X_2 \in 2^{\mathcal{X}}$, $f(X_1) = f(X_2)$ if and only if the two sets $X_1$ and $X_2$ are equal.

We remark that the problem of finding a universal set representation, at least for finite subsets of a countable universe $\mathcal{X}$, has been fully addressed by Zaheer et al. (2017), using the deep set architecture they propose.

With Definitions 3.3 and 3.4, we are now ready to present our main result on constructing universally expressive graph representations.

**Proposition 3.5.** *For each $p = 1, 2, \ldots$, let $f_p$ be an $O(p)$-invariant universal representation function. Further let $g : 2^{\mathbb{R}^3} \to \mathbb{R}$ be a universal set representation. Then the following function*

$$r(G, \boldsymbol{X}_G) = \text{GNN}\left(\boldsymbol{A}_G, \text{concat}\left[\boldsymbol{X}_G, g\left(\left\{\text{concat}\left[\mu_j \boldsymbol{1}_n, \lambda_j \boldsymbol{1}_n, f_{\mu_j}(\boldsymbol{V}_j)\right]\right\}_{j=1}^K\right)\right]\right) \tag{4}$$

*is a universal representation (by Definition 3.1). Here $n = |\mathcal{V}(G)|$, $((\lambda_1, \mu_1), \ldots, (\lambda_K, \mu_K))$ is the spectrum of $G$, and $\boldsymbol{V}_j \in \mathbb{R}^{n \times \mu_j}$ are the $\mu_j$ mutually orthogonal normalized eigenvectors of $\boldsymbol{L}_G$ corresponding to $\lambda_j$. We denote $\boldsymbol{1}_n$ an all-1 vector of shape $n \times 1$. GNN is a maximally expressive MPNN such as the one proposed in (Xu et al., 2018).*

The proof is also given in Appendix A. By Proposition 3.5, the problem of finding a universal representation of graphs is completely reduced to that of finding $O(p)$-invariant universal representations of point sets (as constructions for other components are already known). Therefore, directly applying existing point cloud networks (such as those we will mention in Section 5) to graph Laplacian eigenspaces following equation (4) immediately results in a fairly large design space of GNNs, and universality of the resulting GNN directly follows from universality of the underlying point cloud network.

One may find that equation (4) takes the form of a node feature augmented MPNN. The observation is made explicit with the following definition.

**Definition 3.6** (Vanilla OGE-Aug). Let $f_p$ be an $O(p)$-invariant universal representation, for each $p = 1, 2, \ldots$, and $g : 2^{\mathbb{R}^3} \to \mathbb{R}$ be a universal set representation. Define $Z : \mathcal{G} \to \bigcup_{n=1}^{\infty} \mathbb{R}^n$ as

$$Z(G) = g\left(\left\{\text{concat}\left[\mu_j \boldsymbol{1}_{|\mathcal{V}(G)|}, \lambda_j \boldsymbol{1}_{|\mathcal{V}(G)|}, f_{\mu_j}(\boldsymbol{V}_j)\right]\right\}_{j=1}^K\right), \tag{5}$$

in which the notations follow Proposition 3.5. For $G \in \mathcal{G}$, $Z(G)$ is called a **vanilla orthogonal group equivariant augmentation**, or **Vanilla OGE-Aug** on $G$.

We end this section by discussing the complexity of computing $Z(G)$. The typical complexity of a universal point cloud network is $n \exp(\tilde{O}(\text{dim}))^1$, where $\text{dim}$ is the coordinate dimension. Thus, the complexity of computing equation (5) is $n \exp(\tilde{O}(\max_j \mu_j))$. Our worst-case complexity (in which $\max_j \mu_j \sim n$) matches that of a typical algorithm for graph isomorphism problem (GI). Nevertheless, real-world graphs usually have $\max_j \mu_j \ll n$, making our method computationally affordable in general.

## 4 INCORPORATING THE STABILITY CONSTRAINT

Proposition 3.5 has theoretically confirmed the possibility of finding universal graph representations with Laplacian eigenvectors, even when the backbone GNN is a (relatively weak) MPNN. Nevertheless, naively applying the network architecture proposed in Proposition 3.5 (or Vanilla OGE-Aug) may not necessarily bring performance gain, due to one important weakness—**instability**. As is mentioned in Section 1, instability refers to the proneness to produce very different outputs as the input undergoes small perturbations. Instability of Vanilla OGE-Aug stems from the fact that it **treats Laplacian eigenspaces of different dimensions separately**. As an example, let $\lambda$ be a $K$-fold eigenvalue of Laplacian $\boldsymbol{L}$, whose $K$ corresponding eigenvectors should be encoded by an $O(K)$-invariant universal representation $f_K$; after a small perturbation on $\boldsymbol{L}$, the $K$-dimensional

---

[1] $\tilde{O}(f(n))$ means a complexity linear in $f(n)$ if ignoring poly-logarithm factors, i.e., $O(\log^k f(n))$.

eigenspace corresponding to $\lambda$ might split into two smaller eigenspaces of dimensions $k_1$ and $k_2$ respectively (i.e., the degeneracy of $\lambda$ is partially lifted), which should be alternatively encoded by $f_{k_1}$ and $f_{k_2}$. Since the functions $f_K$ and $f_{k_1}$ (or $f_{k_2}$) can be very different with $K \neq k_1$, $K \neq k_2$, the output can vary a lot even if the changes in $\boldsymbol{L}$ (or changes in the $K$ eigenvalues and eigenvectors) are small.[2] An important lesson from the above discussion is that a "hard split" of Laplacian eigenvectors into separate eigenspaces can be susceptible to perturbations. Hence, model predictions should **not** absolutely rely on such a "hard split" (especially, not relying on the dimension of each eigenspace) for the sake of stability.

According to equation (5), in Vanilla OGE-Aug there are two occurrences of explicit dependencies on eigenspace dimensions $\mu_j$ ($j = 1, \ldots, K$), namely (i) $\mu_j$ being concatenated as a number, and (ii) a different $f_{\mu_j}$ being used for each value of $\mu_j$. To maintain stability, such dependencies should either be removed, or be replaced by functions not sensitive to the exact eigenspace splitting. Our attempt towards this goal is as follows.

**Definition 4.1** (OGE-Aug). Let $G$ be a graph with $n$ nodes. Let $f$ be an $O(n)$-invariant universal representation function. Define

$$\boldsymbol{V}_j^{\text{smooth}} = \text{concat}\left[\boldsymbol{V}_1\rho(|\lambda_1 - \lambda_j|), \boldsymbol{V}_2\rho(|\lambda_2 - \lambda_j|), \ldots, \boldsymbol{V}_K\rho(|\lambda_K - \lambda_j|)\right], \tag{6}$$

where $\rho : \mathbb{R}_{\geqslant 0} \to [0, 1]$ is a continuous *smoothing function* with $\rho(0) = 1$ and $\lim_{x \to +\infty} \rho(x) = 0$, and other notations follow Proposition 3.5. Further let $\phi : \mathbb{R}^2 \to \mathbb{R}^m$ and $\psi : \mathbb{R}^m \to \mathbb{R}$ be parameterized functions that apply row-wise on $n \times 2$ and $n \times m$ matrices, respectively. Then

$$Z(G) = \psi\left(\sum_{j=1}^K \mu_j \phi\left(\text{concat}\left[\lambda_j \mathbf{1}_n, f(\boldsymbol{V}_j^{\text{smooth}})\right]\right)\right) \tag{7}$$

is called an **orthogonal group equivariant augmentation**, or **OGE-Aug** on $G$.

There are some remarkable points regarding OGE-Aug. First, instead of using a different orthogonal group invariant encoder for different eigenspace dimensions, a **single** $O(n)$-invariant encoder $f$ is used to encode eigenvectors coming from all eigenspaces. The dependency on eigenspace dimensions $\mu_j$ ($j = 1, \ldots, K$) appears only in the form of a weighted sum, which is insensitive to the exact splitting of Laplacian eigenspaces. Moreover, a continuous smoothing function $\rho$ is used to keep the eigenvectors aware of the eigenspace where they belong, **as well as the eigenspaces nearby**. As $\rho$ becomes more and more centered at 0 (namely, $\rho(0) = 1$ and $\rho(x) \to 0$ for all $x > 0$), each eigenspace gets encoded by its own portion of parameters from $f$ that are not shared with each other; contrarily, with $\rho$ being flatter, more parameters are shared across eigenspaces. In other words, the shape of $\rho$ controls the "degree of smoothness" of OGE-Aug.

Next, we quantitatively characterize the stability of OGE-Aug. To this end, we first present our definition of stability, following (though slightly different from) (Huang et al., 2024).

**Definition 4.2** (Stability, following Definition 3.1 of (Huang et al., 2024)). A function $f$, operating on the Laplacian $\boldsymbol{L}$ of a graph $G$ and producing a node feature augmentation $\boldsymbol{Z} \in \mathbb{R}^{|\mathcal{V}(G)| \times d}$, is said to be stable, if there exist constants $c_1, C_1, \ldots, c_m, C_m > 0$, such that for any two Laplacians $\boldsymbol{L}, \boldsymbol{L}'$,

$$\|f(\boldsymbol{L}) - \boldsymbol{P}_* f(\boldsymbol{L}')\|_{\text{F}} \leqslant \max_{\ell=1,\ldots,m}\left\{C_\ell \cdot \|\boldsymbol{L} - \boldsymbol{P}_* \boldsymbol{L}' \boldsymbol{P}_*^T\|_{\text{F}}^{c_\ell}\right\}, \tag{8}$$

in which $\|\cdot\|_{\text{F}}$ stands for Frobenius norm, and $\boldsymbol{P}_* = \arg\min_{\boldsymbol{P} \in S_n} \|\boldsymbol{L} - \boldsymbol{P}\boldsymbol{L}'\boldsymbol{P}^T\|_{\text{F}}$ is the permutation matrix matching $\boldsymbol{L}$ and $\boldsymbol{L}'$ (assuming both $\boldsymbol{L}$ and $\boldsymbol{L}'$ are of size $n \times n$).

We are now ready to give our theoretical result on the stability of OGE-Aug. We assume that the following conditions hold for functions $\psi, \phi, f$ and $\rho$.

1. $\psi, \phi$ and $\rho$ are Lipschitz continuous, with Lipschitz constants $J_\psi, J_\phi$ and $J_\rho$ respectively. Namely,

$$\|\psi(\boldsymbol{X}) - \psi(\boldsymbol{X}')\|_{\text{F}} \leqslant J_\psi \|\boldsymbol{X} - \boldsymbol{X}'\|_{\text{F}}, \quad \forall \boldsymbol{X}, \boldsymbol{X}' \in \mathbb{R}^{n \times m}, \tag{9}$$

$$\|\phi(\boldsymbol{X}) - \phi(\boldsymbol{X}')\|_{\text{F}} \leqslant J_\phi \|\boldsymbol{X} - \boldsymbol{X}'\|_{\text{F}}, \quad \forall \boldsymbol{X}, \boldsymbol{X}' \in \mathbb{R}^{n \times 2}, \tag{10}$$

$$|\rho(x) - \rho(x')| \leqslant J_\rho |x - x'|, \quad \forall x, x' \in \mathbb{R}_{\geqslant 0}. \tag{11}$$

---

[2] We remark that a similar problem pertains to BasisNet (Lim et al., 2022). See the discussion in Appendix C of Huang et al. (2024).

2. $f$ satisfies the following condition: $\exists J_f > 0$,

$$\|f(\boldsymbol{X}) - f(\boldsymbol{X}')\| \leqslant J_f \min_{\boldsymbol{Q} \in O(n)} \|\boldsymbol{X} - \boldsymbol{X}'\boldsymbol{Q}\|_{\mathrm{F}}, \quad \forall \boldsymbol{X}, \boldsymbol{X}' \in \mathbb{R}^{n \times n}. \tag{12}$$

One may think of $f$ as $J_f$-Lipschitz continuous after rotating its arguments along the same direction.

3. There exists a constant $\delta > 0$, such that $\rho(x) = 0$ for all $x > \delta$.

Given the above assumptions, we have

**Proposition 4.3** (Stability of OGE-Aug). *With the assumptions on $\psi, \phi, f$ and $\rho$ specified above, OGE-Aug defined by (7) is stable. To be specific, given two graphs $G, G' \in \mathcal{G}$ with Laplacians $\boldsymbol{L}$ and $\boldsymbol{L}'$ respectively, there exists a proper value of $\delta$ such that*

$$\|Z(G) - \boldsymbol{P}_* Z(G')\|_F \leqslant n J_\psi J_\phi \big[ (\sqrt{n} + 2n J_\rho J_f) \|\boldsymbol{L} - \boldsymbol{P}_* \boldsymbol{L}' \boldsymbol{P}_*^T\|_2$$
$$+ 4\sqrt[4]{2} J_f \sqrt{J_\rho} n \|\boldsymbol{L} - \boldsymbol{P}_* \boldsymbol{L}' \boldsymbol{P}_*^T\|_F^{1/2} \big], \tag{13}$$

*where $\|\cdot\|_2$ is the spectral norm which is no larger than the Frobenius norm $\|\cdot\|_F$, and $n = |\mathcal{V}(G)| = |\mathcal{V}(G')|$.*

We give the proof in Appendix B. To ensure that the inequality (13) holds, in principle we need to tune $\delta$ for different $G$ and $G'$. However, in our experiments we simply take $\delta$ as a hyperparameter designated before actual training.

Finally, we discuss practical implementations of OGE-Aug. While presenting the universality result (Proposition 3.5), we have assumed that $f_p$ ($p = 1, 2, \ldots$) can universally represent all $O(p)$-invariant and permutation-equivariant functions on point sets embedded in $\mathbb{R}^p$. This universality requirement is inherited to OGE-Aug (Definition 4.1). Namely, we still require that $f$ is an $O(n)$-invariant universal representation. We now point out that such universality requirement, despite producing maximally expressive networks in theory, can be impractical to implement. First, with $f$ being universal, the resulting network architecture has a typical complexity of $n \exp(\tilde{O}(n))$ which is generally unacceptable. Moreover, insisting on the universality of $f$ can be harmful to the stability of OGE-Aug, since a more expressive $f$ might result in a larger Lipschitz constant $J_f$. Therefore, in our actual implementation of OGE-Aug, we no longer require $f$ to be universal. Instead, we adopt as $f$ a Cartesian tensor based point cloud network (Finkelshtein et al., 2022) with Cartesian tensors up to the second order used. We include more experimental details, as well as a complexity analysis for our implementation, in Appendix D.

## 5 RELATED WORKS

**Graph representation learning with Laplacian eigenvectors.** It is well-known that eigenvectors of graph Laplacian corresponding to the smallest eigenvalues contain "positional" information of nodes. A number of works have thus adopted Laplacian eigenvectors as a technique for node feature augmentation. As we have mentioned in Section 1, there are two important issues regarding the application of Laplacian eigenvectors in graph representation learning, namely orthogonal group invariance (or sign-and-basis invariance) and stability. Some early works (Dwivedi & Bresson, 2020; Kreuzer et al., 2021) have noticed the sign invariance problem and tried to alleviate it by randomly flipping the signs of Laplacian eigenvectors, while completely ignored the basis invariance problem. Lim et al. (2022) is the first work to formally state and systematically address the sign-and-basis invariance issue. Nevertheless, it fails to meet the stronger requirement of stability. So far, only two works (Wang et al., 2022; Huang et al., 2024) have seriously discussed the stability issue by giving mathematical definitions for it, and proposing learning methods that are provably stable.

**Orthogonal-group invariant networks.** A neural network is said to be orthogonal-group invariant if it takes as input one or more vector(s) (say, for instance, each of dimension $p$), and outputs an $O(p)$-invariant scalar, i.e., a value that remains invariant as the input vector system undergoes an $O(p)$ transformation. As is pointed out by, e.g., Bronstein et al. (2021), orthogonal-group invariance is a desirable property for learning tasks on molecular data or point clouds, in which Euclidean coordinates play important roles.

Orthogonal-group equivariance is a property closely related to invariance. A network is $O(p)$-equivariant if it takes as input one (or a set of) arbitrary representation(s)[3] of $O(p)$ (with $p$-dimensional vectors being a special case), and outputs another (or another set of) representation(s) of $O(p)$, in a way that whenever the input system undergoes the action of an $O(p)$ group element, the output also undergoes an action corresponding to the same element. In practice, invariant networks are usually constructed by stacking multiple equivariant layers, along with a final invariant layer. Regarding the intermediate orthogonal group representations they use, existing works on the design of invariant networks mainly take one of the four approaches: (i) utilizing scalar or vector representations (Deng et al., 2021; Li et al., 2024; Satorras et al., 2021; Villar et al., 2021); (ii) utilizing hand-crafted higher-order representations (Gasteiger et al., 2020; 2021; Schütt et al., 2021); (iii) utilizing higher-order Cartesian tensor representations (Finkelshtein et al., 2022; Ruhe et al., 2024); (iv) utilizing higher-order irreducible representations (Batzner et al., 2022; Bogatskiy et al., 2020; Cohen et al., 2018; Fuchs et al., 2020; Thomas et al., 2018).

Similar to the question of expressive power of GNNs, there exists the question of whether an orthogonal-group invariant network can express all possible geometric configurations (either of a single vector or of a point cloud) up to an arbitrary orthogonal transformation. Invariant networks possessing the above property are usually called *universal*. There have been a few works establishing theoretically the universality of some of the aforementioned architectures. Villar et al. (2021) shows that universality can be achieved merely using scalar and vector representations, as long as interaction terms including sufficiently many vectors are allowed, and that the network output is restricted to be scalars or vectors. Li et al. (2024) further shows by construction that an invariant network can be already universal with 4-vector interaction terms, even if all intermediate representations are restricted scalar. Regarding methods using higher-order representations, Dym & Maron (2020) proves the universality of two specific architectures exploiting higher-order irreducible representations of SO(3)—Tensor Field Networks (TFN) (Thomas et al., 2018) and SE(3)-Transformers (Fuchs et al., 2020). Based on TFN, Finkelshtein et al. (2022) proposes another universal architecture utilizing Cartesian tensor representations. The universality results reviewed above have laid theoretical foundations for our proposed method.

We leave the discussion on more related works to Appendix C.

## 6 EXPERIMENTS

In this section, we conduct extensive experiments to evaluate the performance of our methods. We adopt several popular real-world datasets, including: (1) QM9 (Ramakrishnan et al., 2014); (2) ZINC12k (Dwivedi et al., 2020); (3) Alchemy (Chen et al., 2019); (4) PCQM-Contact (Dwivedi et al., 2022); (5) CLUSTER (Dwivedi et al., 2023); (6) PATTERN (Dwivedi et al., 2023); (7) ogbg-molhiv (Hu et al., 2021); (8) DrugOOD (Ji et al., 2022). Results on the first four datasets are given below, while other experimental results are given in Appendix D. Dataset statistics are summarized in Table 5. We also provide detailed experimental settings in Appendix D.

**QM9.** QM9 (Ramakrishnan et al., 2014) is a graph property regression dataset containing 130k small molecules and 19 regression targets. We use a commonly adopted 0.8/0.1/0.1 training/validation/test split ratio, and report the results of the first 12 targets. Several representative expressive GNNs are selected as baselines, including MPNN, 1-2-3-GNN (Morris et al., 2019), DTNN (Wu et al., 2017), DeepLRP (Chen et al., 2020), PPGN (Maron et al., 2019), NGNN (Zhang & Li, 2021), KP-GIN+ (Feng et al., 2022), IDMPNN (Zhou et al., 2023b) and PST (Wang et al., 2024). The results are shown in Table 1. From Table 1, we find that OGE-Aug achieves competitive performance on all 12 targets. We also notice that our method achieves a relatively low MAE on targets $U_0$, $U$, $H$ and $G$, compared with subtree- or subgraph-based methods such as MPNN, NGNN or KP-GIN+, as well as other Laplacian eigenvector augmented GNNs like PST. This fact indicates that our method has the ability to capture **global properties** of graphs, since those targets are macroscopic thermodynamic properties of molecules and heavily depend on long-range interactions (for example, intermolecular forces like hydrogen bonds).

---

[3]In our context, a representation of $O(p)$ means a vector lying in a linear space $\mathcal{L}$, given that a group homomorphism from $O(p)$ to the general linear group $\mathrm{GL}(\mathcal{L})$ on $\mathcal{L}$ exists.

Table 1: QM9 results (MAE ↓). Highlighted are **first**, **second** best results.

| Target | MPNN | 1-2-3-GNN | DTNN | DeepLRP | PPGN | NGNN | KP-GIN+ | 4-IDMPNN | PST | OGE-Aug |
|---|---|---|---|---|---|---|---|---|---|---|
| $\mu$ | 0.358 | 0.476 | 0.244 | 0.364 | 0.231 | 0.433 | 0.358 | 0.398 | **0.023** | **0.0822** |
| $\alpha$ | 0.89 | 0.27 | 0.95 | 0.298 | 0.382 | 0.265 | 0.233 | 0.226 | **0.078** | **0.159** |
| $\epsilon_{\text{HOMO}}$ | 0.00541 | 0.00337 | 0.00388 | 0.00254 | 0.00276 | 0.00279 | 0.00240 | 0.00263 | **0.00110** | **0.00140** |
| $\epsilon_{\text{LUMO}}$ | 0.00623 | 0.00351 | 0.00512 | 0.00277 | 0.00287 | 0.00276 | 0.00236 | 0.00286 | **0.00081** | **0.00144** |
| $\Delta\epsilon$ | 0.0066 | 0.0048 | 0.0112 | 0.00353 | 0.00406 | 0.00390 | 0.00333 | 0.00398 | **0.0016** | **0.00198** |
| $\langle R^2 \rangle$ | 28.5 | 22.9 | 17.0 | 19.3 | 16.7 | 20.1 | 16.49 | 10.4 | **0.93** | **5.55** |
| ZPVE | 0.00216 | 0.00019 | 0.00172 | 0.00055 | 0.00064 | 0.00015 | 0.00017 | **0.00013** | **0.000095** | 0.000149 |
| $U_0$ | 2.05 | **0.0427** | 2.43 | 0.413 | 0.234 | 0.205 | 0.0682 | **0.0189** | 0.121 | 0.0526 |
| $U$ | 2.00 | 0.111 | 2.43 | 0.413 | 0.234 | 0.200 | 0.0553 | **0.0152** | 0.120 | **0.0356** |
| $H$ | 2.02 | **0.0419** | 2.43 | 0.413 | 0.229 | 0.249 | 0.0575 | **0.0160** | 0.118 | 0.0439 |
| $G$ | 2.02 | 0.0469 | 2.43 | 0.413 | 0.238 | 0.253 | 0.0484 | **0.0159** | 0.119 | **0.0441** |
| $c_{\text{v}}$ | 0.42 | 0.0944 | 0.27 | 0.129 | 0.184 | 0.0811 | 0.0869 | 0.0890 | **0.0363** | **0.0681** |

**ZINC.** ZINC12k (Dwivedi et al., 2020) is a subset of the ZINC250k dataset containing 12k molecules, and the task is molecular property (constrained solubility) regression evaluated by mean absolute error (MAE). We follow the official split of the dataset. We include common baselines such as GIN (Xu et al., 2018), PNA (Corso et al., 2020), DeepLRP (Chen et al., 2020), OSAN (Qian et al., 2022), KP-GIN+ (Feng et al., 2022), GNN-AK+ (Zhao et al., 2021) and CIN (Bodnar et al., 2021).

We also include previous methods making use of Laplacian eigenvectors to produce node feature augmentations (which are usually named *positional encodings* or PEs), such as PEG (Wang et al., 2022), SignNet (Lim et al., 2022), Basis-Net (Lim et al., 2022) and SPE (Huang et al., 2024), as well as graph transformers such as SAN (Kreuzer et al., 2021), Graphormer (Ying et al., 2021a), GraphGPS (Rampášek et al., 2022) and Specformer (Bo et al., 2023). Among the graph transformer baselines, SAN, GraphGPS and Specformer also encode spectral information through other approaches. Regarding our OGE-Aug, we consider both GINE (Hu et al., 2019) (which belongs to the MPNN family) and the GPS as base models. As shown in Table 2, OGE-Aug outperforms all baseline methods even combined with the simple GINE backbone without global attention.

**Alchemy.** Alchemy (Chen et al., 2019) is also a graph-level small molecular property regression dataset from the TUDatasets. We adopt message-passing GNN backbones, and consider alternative expressive PEs including PEG (Wang et al., 2022), SignNet (Lim et al., 2022), BasisNet (Lim et al., 2022) and SPE (Huang et al., 2024). As shown in Table 3, our OGE-Aug significantly outperforms all these baselines and achieves state-of-the-art performance.

Table 2: Zinc12K results (MAE ↓). Shown is the mean ± std of 5 runs.

| Method | Test MAE |
|---|---|
| GIN | $0.163 \pm 0.004$ |
| PNA | $0.188 \pm 0.004$ |
| GSN | $0.115 \pm 0.012$ |
| OSAN | $0.187 \pm 0.004$ |
| KP-GIN+ | $0.119 \pm 0.002$ |
| GNN-AK+ | $0.080 \pm 0.001$ |
| CIN | $0.079 \pm 0.006$ |
| GIN, with PEG | $0.144 \pm 0.008$ |
| GIN, with SignNet | $0.085 \pm 0.003$ |
| GIN, with BasisNet | $0.155 \pm 0.007$ |
| GIN, with SPE | $0.069 \pm 0.004$ |
| SAN | $0.139 \pm 0.006$ |
| Graphormer | $0.122 \pm 0.006$ |
| GPS | $0.070 \pm 0.004$ |
| Specformer | $0.066 \pm 0.003$ |
| GINE, with OGE-Aug (ours) | $0.066 \pm 0.002$ |
| GPS, with OGE-Aug (ours) | $\mathbf{0.064 \pm 0.003}$ |

Table 3: Experiments on Alchemy. Shown is the mean ± std of 5 runs with different random seeds.

| Model | PE | Test MAE ↓ |
|---|---|---|
| GIN | None | $0.112 \pm 0.001$ |
| GIN | PEG (8) | $0.114 \pm 0.001$ |
| GIN | SignNet (All) | $0.113 \pm 0.002$ |
| GIN | BasisNet (All) | $0.110 \pm 0.001$ |
| GIN | SPE (All) | $0.108 \pm 0.001$ |
| GINE | OGE-Aug (ours) | $\mathbf{0.087 \pm 0.001}$ |

Table 4: Experiments on PCQM-Contact dataset from the long-range graph benchmarks (LRGB). Highlighted are the first, second, third best results.

| model | PE | PCQM-Contact (MRR ↑) |
|---|---|---|
| GCN | None | $0.3234 \pm 0.0006$ |
| GINE | None | $0.3180 \pm 0.0027$ |
| GatedGCN | None | $0.3218 \pm 0.0011$ |
| Transformer | LapPE | $0.3174 \pm 0.0020$ |
| SAN | LapPE | $0.3350 \pm 0.0003$ |
| SAN | RWSE | $0.3341 \pm 0.0006$ |
| GPS | LapPE | $0.3337 \pm 0.0006$ |
| GPS | EdgeRWSE | $\mathbf{0.3408 \pm 0.0003}$ |
| GPS | Hodge1Lap | $0.3407 \pm 0.0004$ |
| Exphormer | None | $\mathbf{0.3637 \pm 0.0020}$ |
| GPS | OGE-Aug (ours) | $\mathbf{0.3543 \pm 0.0004}$ |

**PCQM-Contact.** As part of the long-range graph benchmarks (LRGB) (Dwivedi et al., 2022), PCQM-Contact is a dataset derived from the PCQM4Mv2 dataset along with the corresponding 3D molecular structures. The task is a binary link ranking measured by the Mean Reciprocal Rank (MRR), which requires the capability of capturing long range interactions. MPNN baselines include GCN (Kipf & Welling, 2016), GINE (Hu et al., 2019), and GatedGCN (Bresson & Laurent, 2017), while graph transformer baselines include Transformer, SAN, Exphormer (Shirzad et al., 2023) and GPS combined with positional encodings (PEs) like LapPE (Kreuzer et al., 2021), RWSE (Dwivedi et al., 2021), EdgeRWSE (Zhou et al., 2023a) and Hodge1Lap (Zhou et al., 2023a). We combine GPS with our OGE-Aug and achieve the second best performance across all baselines, which verifies the benefit of bringing in long-range information via OGE-Aug.

## 7 CONCLUSION

In this paper, we propose to apply orthogonal group invariant neural networks on Laplacian eigenspaces of graphs, so as to produce node feature augmentations that may possess great expressive power. We present Vanilla OGE-Aug and OGE-Aug as two instances of our proposed framework, of which the former illustrates the potential of our method to achieve universal representation of graphs, while the latter is provably stable and practically useful. Extensive experiments have verified the outstanding performance of OGE-Aug on various benchmarks as well as its capability to learn global properties of graphs. We remark that our approach to incorporating stability into graph learning methods based on Laplacian eigenvectors, i.e., by ensuring *smoothness* while processing different Laplacian eigenspaces, is a general technique, and can be applied to other machine learning domains where eigenvalues and eigenvectors are of significant interest.

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

## A    PROOFS OF PROPOSITIONS IN SECTION 3

### A.1    PROOF OF PROPOSITION 3.2

*Proof.* We denote $n = |\mathcal{V}(G)|$. Let $\boldsymbol{L}_G$ and $\boldsymbol{L}_H$ be the Laplacians of $G$ and $H$, respectively. We first show that statement (i) is equivalent to the following: $\exists \boldsymbol{P} \in S_n, \boldsymbol{L}_G = \boldsymbol{P}\boldsymbol{L}_H\boldsymbol{P}^T$. By definition of permutation matrices, for any $\boldsymbol{P} \in S_n$ there exists a bijective function $p : \{1, \dots, n\} \to \{1, \dots, n\}$ such that $P_{ij} = 1_{p(i)=j}$. Therefore, we have $\boldsymbol{L}_G = \boldsymbol{P}\boldsymbol{L}_H\boldsymbol{P}^T \Leftrightarrow L_{Gij} = L_{Hp(i)p(j)}$. Since the off-diagonal part of $\boldsymbol{L}_G$ (or $\boldsymbol{L}_H$) is $-\boldsymbol{A}_G$ (or $-\boldsymbol{A}_H$), $L_{Gij} = L_{Hp(i)p(j)}$ implies $A_{Gij} = A_{Hp(i)p(j)}$. Thus $\boldsymbol{A}_G = \boldsymbol{P}\boldsymbol{A}_H\boldsymbol{P}^T$ follows from $\boldsymbol{L}_G = \boldsymbol{P}\boldsymbol{L}_H\boldsymbol{P}^T$. To see the other direction, notice that given $\boldsymbol{A}_G = \boldsymbol{P}\boldsymbol{A}_H\boldsymbol{P}^T$ or $A_{Gij} = A_{Hp(i)p(j)}$, we have

$$(\boldsymbol{P}\boldsymbol{D}_H\boldsymbol{P}^T)_{ij} = D_{Hp(i)p(i)}1_{i=j} = \sum_{k=1}^{n} A_{Hp(i)p(k)}1_{i=j} = \sum_{k=1}^{n} A_{Gik}1_{i=j} = D_{Gij}, \quad (14)$$

or simply $\boldsymbol{D}_G = \boldsymbol{P}\boldsymbol{D}_H\boldsymbol{P}^T$. Thus $\boldsymbol{L}_G = \boldsymbol{P}\boldsymbol{L}_H\boldsymbol{P}^T$.

Next, we prove that statement (ii) is equivalent to $\exists \boldsymbol{P} \in S_n, \boldsymbol{L}_G = \boldsymbol{P}\boldsymbol{L}_H\boldsymbol{P}^T$. Assuming that statement (ii) is true, one may make use of the identities

$$\boldsymbol{L}_G = \sum_{j=1}^{K} \lambda_j \boldsymbol{V}_j \boldsymbol{V}_j^T, \quad \boldsymbol{L}_H = \sum_{j=1}^{K} \lambda_j \boldsymbol{V}_j' \boldsymbol{V}_j'^T, \quad (15)$$

to observe that $\boldsymbol{L}_G = \boldsymbol{P}\boldsymbol{L}_H\boldsymbol{P}^T$. To see the other direction, notice that $\boldsymbol{L}_G = \boldsymbol{P}\boldsymbol{L}_H\boldsymbol{P}^T$ implies that $\boldsymbol{L}_G$ and $\boldsymbol{L}_H$ are similar, and thus $\mathrm{Spec}\, G = \mathrm{Spec}\, H$ as similar matrices share the set of eigenvalues combined with their corresponding multiplicities. Moreover, if the columns of $\boldsymbol{V}_j'$ constitute the set of mutually orthogonal normalized eigenvectors of $\boldsymbol{L}_H$ corresponding to eigenvalue $\lambda_j$, then the columns of $\boldsymbol{P}\boldsymbol{V}_j'$ contain mutually orthogonal normalized eigenvectors of $\boldsymbol{L}_G$ corresponding to the same eigenvalue, for $j = 1, \dots, K$. Therefore, each column of $\boldsymbol{V}_j$ must be a linear combination of columns of $\boldsymbol{P}\boldsymbol{V}_j'$, namely

$$\boldsymbol{V}_j = \boldsymbol{P}\boldsymbol{V}_j'\boldsymbol{Q}_j, \quad (16)$$

for some $\boldsymbol{Q}_j \in \mathbb{R}^{\mu_j \times \mu_j}$. Further imposing the constraint that $\boldsymbol{V}_j^T\boldsymbol{V}_j = \boldsymbol{I}_{\mu_j \times \mu_j}$ yields $\boldsymbol{Q}_j^T\boldsymbol{Q}_j = \boldsymbol{I}_{\mu_j \times \mu_j}$, or $\boldsymbol{Q}_j \in O(\mu_j)$. Thus the proof is made. $\qquad \square$

### A.2    PROOF OF PROPOSITION 3.5

*Proof.* By Definition 3.1, we only need to prove that $r(G, \boldsymbol{X}_G) = r(H, \boldsymbol{X}_H)$ if and only if $\exists \boldsymbol{P} \in S_n$ such that $\boldsymbol{A}_G = \boldsymbol{P}\boldsymbol{A}_H\boldsymbol{P}^T$ and $\boldsymbol{X}_G = \boldsymbol{P}\boldsymbol{X}_H$, for any two graphs $G, H$ with accompanying node features $\boldsymbol{X}_G, \boldsymbol{X}_H$. By Proposition 3.2, the latter condition is equivalent to the conjunction of the following:

1. $\mathrm{Spec}\, G = \mathrm{Spec}\, H$.

2. $\exists \boldsymbol{P} \in S_n$ and $\boldsymbol{Q}_j \in O(\mu_j)$ $(j = 1, \dots, K)$, such that $\boldsymbol{X}_G = \boldsymbol{P}\boldsymbol{X}_H$, and $\boldsymbol{V}_j = \boldsymbol{P}\boldsymbol{V}_j'\boldsymbol{Q}_j$, for $j = 1, \dots, K$.

Our notations follow those in Proposition 3.2. Now, given that the above two conditions are true, we immediately get $f_{\mu_j}(\boldsymbol{V}_j) = \boldsymbol{P}f_{\mu_j}(\boldsymbol{V}_j')$ due to the fact that $f_{\mu_j}$ is an $O(\mu_j)$-invariant universal representation. Thus, we have

$$\mathrm{concat}\left[\mu_j\boldsymbol{1}_n, \lambda_j\boldsymbol{1}_n, f_{\mu_j}(\boldsymbol{V}_j)\right] = \boldsymbol{P}\,\mathrm{concat}\left[\mu_j\boldsymbol{1}_n, \lambda_j\boldsymbol{1}_n, f_{\mu_j}(\boldsymbol{V}_j')\right]. \quad (17)$$

Similarly, since $g$ operates on individual rows of set elements, the permutation matrix $\boldsymbol{P}$ passes through the operation of $g$. Therefore,

$$g\left(\left\{\mathrm{concat}\left[\mu_j\boldsymbol{1}_n, \lambda_j\boldsymbol{1}_n, f_{\mu_j}(\boldsymbol{V}_j)\right]\right\}_{j=1}^{K}\right) = \boldsymbol{P}g\left(\left\{\mathrm{concat}\left[\mu_j\boldsymbol{1}_n, \lambda_j\boldsymbol{1}_n, f_{\mu_j}(\boldsymbol{V}_j')\right]\right\}_{j=1}^{K}\right). \quad (18)$$

If we let

$$X'_G = \text{concat}\left[X_G, g\left(\left\{\text{concat}\left[\mu_j \mathbf{1}_n, \lambda_j \mathbf{1}_n, f_{\mu_j}(V_j)\right]\right\}_{j=1}^K\right)\right], \tag{19}$$

$$X'_H = \text{concat}\left[X_H, g\left(\left\{\text{concat}\left[\mu_j \mathbf{1}_n, \lambda_j \mathbf{1}_n, f_{\mu_j}(V'_j)\right]\right\}_{j=1}^K\right)\right], \tag{20}$$

then $X'_G = P X'_H$. Since message passing GNNs are invariant with respect to node permutations, we know that

$$r(G, X_G) = \text{GNN}(A_G, X'_G) = \text{GNN}(P A_H P^T, P X'_H) = \text{GNN}(A_H, X'_H) = r(H, X_H), \tag{21}$$

thus proving one direction of the proposition.

For the other direction, notice that a maximally expressive message passing GNN is as powerful as the 1-WL test (Xu et al., 2018), and strictly stronger than a universal set encoder (regarding the set of node features).[4] Therefore, by construction (4), $r(G, X_G) = r(H, X_H)$ implies that $\exists P \in S_n$, $X'_G = P X'_H$, where $X'_G$ is defined in equation (19) but $X'_H$ should be alternatively defined as

$$X'_H = \text{concat}\left[X_H, g\left(\left\{\text{concat}\left[\mu'_j \mathbf{1}_n, \lambda'_j \mathbf{1}_n, f_{\mu'_j}(V'_j)\right]\right\}_{j=1}^{K'}\right)\right], \tag{22}$$

since we have not yet proved that $G$ and $H$ share spectra. The above fact further translates into $X_G = P X_H$ and

$$g\left(\left\{\text{concat}\left[\mu_j \mathbf{1}_n, \lambda_j \mathbf{1}_n, f_{\mu_j}(V_j)\right]\right\}_{j=1}^K\right) = P g\left(\left\{\text{concat}\left[\mu'_j \mathbf{1}_n, \lambda'_j \mathbf{1}_n, f_{\mu'_j}(V'_j)\right]\right\}_{j=1}^{K'}\right). \tag{23}$$

Since $g$ is a universal set representation, the sets on both sides are equal up to an element-wise application of $P$. As a consequence,

$$\{(\mu_j, \lambda_j)\}_{j=1}^K = \{(\mu'_j, \lambda'_j)\}_{j=1}^{K'}, \tag{24}$$

or $\text{Spec } G = \text{Spec } H$. Now that $G$ and $H$ share spectra, we may assume that the eigenvalues $\{\lambda_j\}_{j=1}^K$ are in an order such that $0 = \lambda_1 < \lambda_2 < \cdots < \lambda_K$. We then arrive at equation (17), and subsequently $f_{\mu_j}(V_j) = P f_{\mu_j}(V'_j)$, for each $j = 1, \ldots, K$. Due to $f_{\mu_j}$ being an $O(\mu_j)$-invariant universal representation, we end up finding that $\exists Q_j \in O(\mu_j)$ $(j = 1, \ldots, K)$, such that $V_j = P V'_j Q_j$. So far we have proved the other direction of the proposition. $\square$

## B  PROOF OF PROPOSITION 4.3

Before proving Proposition 4.3, we present some useful lemmas. We quote these lemmas directly from (Huang et al., 2024).

**Lemma B.1** (Davis-Kahan theorem, Proposition A.1 of (Huang et al., 2024), see also (Yu et al., 2015))**.** *Let $A, A'$ be $n \times n$ real symmetric matrices. Let $\lambda_1 \leqslant \cdots \leqslant \lambda_n$ be eigenvalues of $A$ sorted in increasing order (possibly with repeats). Let the columns of $V, V' \in O(n)$ contain mutually orthogonal normalized eigenvectors of $A, A'$ respectively, sorted in increasing order of their corresponding eigenvalues. Let $\mathcal{J} = \{s, s+1, \ldots, t\} \subseteq \{1, \ldots, n\}$ be a contiguous interval of indices, and $[V]_{\mathcal{J}}, [V']_{\mathcal{J}}$ be matrices of shape $n \times |\mathcal{J}|$ whose columns are the s-th, $(s+1)$-th, ..., t-th column of $V$ and $V'$, respectively. Then*

$$\min_{Q \in O(|\mathcal{J}|)} \|[V]_{\mathcal{J}} - [V']_{\mathcal{J}} Q\|_F \leqslant \frac{\sqrt{8} \min\left\{\sqrt{|\mathcal{J}|} \|A - A'\|_2, \|A - A'\|_F\right\}}{\min\{\lambda_s - \lambda_{s-1}, \lambda_{t+1} - \lambda_t\}}. \tag{25}$$

*For convenience, we define $\lambda_0 = -\infty$ and $\lambda_{n+1} = +\infty$.*

**Lemma B.2** (Weyl's inequality, Proposition A.2 of (Huang et al., 2024))**.** *Given a real symmetric matrix $A$, let $\lambda_i(A)$ be its i-th smallest eigenvalue. For any two real symmetric matrices $A, A'$ of shape $n \times n$, $|\lambda_i(A) - \lambda_i(A')| \leqslant \|A - A'\|_2$ holds for all $i = 1, \ldots, n$.*

---

[4]Indeed, a message passing GNN with a maximally expressive pooling layer and no message passing layers is equivalent to a deep set, the latter having proved to be a universal set encoder by Zaheer et al. (2017).

**Lemma B.3** (Lemma A.1 of (Huang et al., 2024)). *Assume $\boldsymbol{A}_1 \boldsymbol{A}_2 \cdots \boldsymbol{A}_p$ is a valid matrix multiplication. Then*

$$\left\| \prod_{k=1}^{p} \boldsymbol{A}_k \right\|_F \leqslant \left( \prod_{k=1}^{\ell-1} \|\boldsymbol{A}_k\|_2 \right) \|\boldsymbol{A}_\ell\|_F \left( \prod_{k=\ell+1}^{p} \|\boldsymbol{A}_k^T\|_2 \right). \tag{26}$$

Now we can present the proof of Proposition 4.3.

*Proof.* We will prove the uniform result that for any two graphs $G, G' \in \mathcal{G}$ with Laplacians $\boldsymbol{L}, \boldsymbol{L}'$ respectively, and for any $\boldsymbol{P} \in S_n$, there exists a value of $\delta$ such that

$$\|Z(G) - \boldsymbol{P}Z(G')\|_F \leqslant n J_\psi J_\phi \big[ (\sqrt{n} + 2n J_\rho J_f) \|\boldsymbol{L} - \boldsymbol{P}\boldsymbol{L}'\boldsymbol{P}^T\|_2$$
$$+ 4\sqrt[4]{2} J_f \sqrt{J_\rho} n \|\boldsymbol{L} - \boldsymbol{P}\boldsymbol{L}'\boldsymbol{P}^T\|_F^{1/2} \big]. \tag{27}$$

We may first rewrite equation (7) as

$$Z(G) = \psi \left( \sum_{i=1}^{n} \phi \left( \mathrm{concat} \left[ \tilde{\lambda}_i \mathbf{1}_n, f(\tilde{\boldsymbol{V}}_i^{\mathrm{smooth}}) \right] \right) \right), \tag{28}$$

in which $\tilde{\lambda}_i$ is the $i$-th smallest eigenvalue of $\boldsymbol{L}$ (*including repeats* when counting orders), and

$$\tilde{\boldsymbol{V}}_i^{\mathrm{smooth}} = \mathrm{concat} \left[ \boldsymbol{v}_1 \rho(|\tilde{\lambda}_1 - \tilde{\lambda}_i|), \boldsymbol{v}_2 \rho(|\tilde{\lambda}_2 - \tilde{\lambda}_i|), \dots, \boldsymbol{v}_n \rho(|\tilde{\lambda}_n - \tilde{\lambda}_i|) \right], \tag{29}$$

where column vectors $\boldsymbol{v}_1, \boldsymbol{v}_2, \dots, \boldsymbol{v}_n \in \mathbb{R}^{n \times 1}$ are mutually orthogonal normalized eigenvectors corresponding to eigenvalues $\tilde{\lambda}_1, \tilde{\lambda}_2, \dots, \tilde{\lambda}_n$ respectively. With equation (28), we have completely removed the dependency on eigenspace dimensions in $Z(G)$. We then have

$$\|Z(G) - \boldsymbol{P}Z(G')\|_F = \left\| \psi \left( \sum_{i=1}^{n} \phi \left( \mathrm{concat} \left[ \tilde{\lambda}_i \mathbf{1}_n, f(\tilde{\boldsymbol{V}}_i^{\mathrm{smooth}}) \right] \right) \right) \right.$$
$$\left. - \boldsymbol{P} \psi \left( \sum_{i=1}^{n} \phi \left( \mathrm{concat} \left[ \tilde{\lambda}_i' \mathbf{1}_n, f(\tilde{\boldsymbol{V}}_i'^{\mathrm{smooth}}) \right] \right) \right) \right\|_F \tag{30}$$

$$= \left\| \psi \left( \sum_{i=1}^{n} \phi \left( \mathrm{concat} \left[ \tilde{\lambda}_i \mathbf{1}_n, f(\tilde{\boldsymbol{V}}_i^{\mathrm{smooth}}) \right] \right) \right) \right.$$
$$\left. - \psi \left( \sum_{i=1}^{n} \phi \left( \mathrm{concat} \left[ \tilde{\lambda}_i' \mathbf{1}_n, f(\boldsymbol{P}\tilde{\boldsymbol{V}}_i'^{\mathrm{smooth}}) \right] \right) \right) \right\|_F \tag{31}$$

$$\leqslant J_\psi \left\| \sum_{i=1}^{n} \phi \left( \mathrm{concat} \left[ \tilde{\lambda}_i \mathbf{1}_n, f(\tilde{\boldsymbol{V}}_i^{\mathrm{smooth}}) \right] \right) \right.$$
$$\left. - \sum_{i=1}^{n} \phi \left( \mathrm{concat} \left[ \tilde{\lambda}_i' \mathbf{1}_n, f(\boldsymbol{P}\tilde{\boldsymbol{V}}_i'^{\mathrm{smooth}}) \right] \right) \right\|_F \tag{32}$$

$$\leqslant J_\psi \sum_{i=1}^{n} \left\| \phi \left( \mathrm{concat} \left[ \tilde{\lambda}_i \mathbf{1}_n, f(\tilde{\boldsymbol{V}}_i^{\mathrm{smooth}}) \right] \right) \right.$$
$$\left. - \phi \left( \mathrm{concat} \left[ \tilde{\lambda}_i' \mathbf{1}_n, f(\boldsymbol{P}\tilde{\boldsymbol{V}}_i'^{\mathrm{smooth}}) \right] \right) \right\|_F \tag{33}$$

$$\leqslant J_\psi J_\phi \sum_{i=1}^{n} \left\| \mathrm{concat} \left[ \left( \tilde{\lambda}_i - \tilde{\lambda}_i' \right) \mathbf{1}_n, f(\tilde{\boldsymbol{V}}_i^{\mathrm{smooth}}) - f(\boldsymbol{P}\tilde{\boldsymbol{V}}_i'^{\mathrm{smooth}}) \right] \right\|_F \tag{34}$$

$$\leqslant J_\psi J_\phi \sum_{i=1}^{n} \left[ \sqrt{n} \left| \tilde{\lambda}_i - \tilde{\lambda}_i' \right| + \|f(\tilde{\boldsymbol{V}}_i^{\mathrm{smooth}}) - f(\boldsymbol{P}\tilde{\boldsymbol{V}}_i'^{\mathrm{smooth}})\| \right]. \tag{35}$$

The equality on (31) is due to the fact that $\psi$ and $\phi$ operate row-wise on the $n$ rows of their arguments, and that $f$ is permutation equivariant. (32) and (34) stem from the Lipschitz continuities of $\psi$ and $\phi$, respectively. (33) is due to triangular inequality. Now it suffices to bound the two terms in (35).

For the first term, we invoke Lemma B.2 to get

$$\sum_{i=1}^n \left|\tilde{\lambda}_i - \tilde{\lambda}_i'\right| \leqslant \sum_{i=1}^n \|\boldsymbol{L} - \boldsymbol{P}\boldsymbol{L}'\boldsymbol{P}^T\|_2 = n\|\boldsymbol{L} - \boldsymbol{P}\boldsymbol{L}'\boldsymbol{P}^T\|_2, \quad \forall \boldsymbol{P} \in S_n. \tag{36}$$

This is because for any $\boldsymbol{P} \in S_n$, $\boldsymbol{P}\boldsymbol{L}'\boldsymbol{P}^T$ has the same sequence of eigenvalues as $\boldsymbol{L}'$, namely $\tilde{\lambda}_1', \tilde{\lambda}_2', \ldots, \tilde{\lambda}_n'$.

For the second term, we have

$$\|f(\tilde{\boldsymbol{V}}_i^{\text{smooth}}) - f(\boldsymbol{P}\tilde{\boldsymbol{V}}_i'^{\text{smooth}})\| \leqslant J_f \min_{\boldsymbol{Q}_i \in O(n)} \|\tilde{\boldsymbol{V}}_i^{\text{smooth}} - \boldsymbol{P}\tilde{\boldsymbol{V}}_i'^{\text{smooth}}\boldsymbol{Q}_i\|_{\text{F}} \tag{37}$$

$$= J_f \min_{\boldsymbol{Q}_i \in O(n)} \left\|\text{concat}\left[\boldsymbol{v}_1\rho(|\tilde{\lambda}_1 - \tilde{\lambda}_i|), \ldots, \boldsymbol{v}_n\rho(|\tilde{\lambda}_n - \tilde{\lambda}_i|)\right]\right.$$

$$\left. - \text{concat}\left[\boldsymbol{P}\boldsymbol{v}_1'\rho(|\tilde{\lambda}_1' - \tilde{\lambda}_i'|), \ldots, \boldsymbol{P}\boldsymbol{v}_n'\rho(|\tilde{\lambda}_n' - \tilde{\lambda}_i'|)\right]\boldsymbol{Q}_i\right\|_{\text{F}} \tag{38}$$

$$\leqslant J_f \min_{\boldsymbol{Q}_i \in O(n)} \left\{\left\|\text{concat}\left[\boldsymbol{v}_1\rho(|\tilde{\lambda}_1 - \tilde{\lambda}_i|), \ldots, \boldsymbol{v}_n\rho(|\tilde{\lambda}_n - \tilde{\lambda}_i|)\right]\right.\right.$$

$$\left. - \text{concat}\left[\boldsymbol{v}_1\rho(|\tilde{\lambda}_1' - \tilde{\lambda}_i'|), \ldots, \boldsymbol{v}_n\rho(|\tilde{\lambda}_n' - \tilde{\lambda}_i'|)\right]\right\|_{\text{F}}$$

$$+ \left\|\text{concat}\left[\boldsymbol{v}_1\rho(|\tilde{\lambda}_1' - \tilde{\lambda}_i'|), \ldots, \boldsymbol{v}_n\rho(|\tilde{\lambda}_n' - \tilde{\lambda}_i'|)\right]\right.$$

$$\left.\left. - \text{concat}\left[\boldsymbol{P}\boldsymbol{v}_1'\rho(|\tilde{\lambda}_1' - \tilde{\lambda}_i'|), \ldots, \boldsymbol{P}\boldsymbol{v}_n'\rho(|\tilde{\lambda}_n' - \tilde{\lambda}_i'|)\right]\boldsymbol{Q}_i\right\|_{\text{F}}\right\} \tag{39}$$

$$= J_f \sqrt{\sum_{j=1}^n \left[\rho(|\tilde{\lambda}_j - \tilde{\lambda}_i|) - \rho(|\tilde{\lambda}_j' - \tilde{\lambda}_i'|)\right]^2}$$

$$+ J_f \min_{\boldsymbol{Q}_i \in O(n)} \left\|\text{concat}\left[\boldsymbol{v}_1\rho(|\tilde{\lambda}_1' - \tilde{\lambda}_i'|), \ldots, \boldsymbol{v}_n\rho(|\tilde{\lambda}_n' - \tilde{\lambda}_i'|)\right]\right.$$

$$\left. - \text{concat}\left[\boldsymbol{P}\boldsymbol{v}_1'\rho(|\tilde{\lambda}_1' - \tilde{\lambda}_i'|), \ldots, \boldsymbol{P}\boldsymbol{v}_n'\rho(|\tilde{\lambda}_n' - \tilde{\lambda}_i'|)\right]\boldsymbol{Q}_i\right\|_{\text{F}}. \tag{40}$$

Here, (37) is due to our assumption on $f$, (38) follows from definitions of $\tilde{\boldsymbol{V}}_i^{\text{smooth}}$ and $\tilde{\boldsymbol{V}}_i'^{\text{smooth}}$, while (39) stems from triangular inequality. Now, for the first term of (40), we have

$$\sqrt{\sum_{j=1}^n \left[\rho(|\tilde{\lambda}_j - \tilde{\lambda}_i|) - \rho(|\tilde{\lambda}_j' - \tilde{\lambda}_i'|)\right]^2} \leqslant \sum_{j=1}^n \left|\rho(|\tilde{\lambda}_j - \tilde{\lambda}_i|) - \rho(|\tilde{\lambda}_j' - \tilde{\lambda}_i'|)\right| \tag{41}$$

$$\leqslant J_\rho \sum_{j=1}^n \left||\tilde{\lambda}_j - \tilde{\lambda}_i| - |\tilde{\lambda}_j' - \tilde{\lambda}_i'|\right| \tag{42}$$

$$\leqslant J_\rho \sum_{j=1}^n \left(|\tilde{\lambda}_i - \tilde{\lambda}_i'| + |\tilde{\lambda}_j - \tilde{\lambda}_j'|\right) \tag{43}$$

$$\leqslant 2n J_\rho \|\boldsymbol{L} - \boldsymbol{P}\boldsymbol{L}'\boldsymbol{P}^T\|_2, \quad \forall \boldsymbol{P} \in S_n, \tag{44}$$

where (42) is by Lipschitz continuity of $\rho$, (43) makes use of the fact that either $\tilde{\lambda}_i \geqslant \tilde{\lambda}_j$ and $\tilde{\lambda}_i' \geqslant \tilde{\lambda}_j'$, or $\tilde{\lambda}_i \leqslant \tilde{\lambda}_j$ and $\tilde{\lambda}_i' \leqslant \tilde{\lambda}_j'$. The final step (44) stems from Lemma B.2.

To bound the second term of (40), we first split the eigenvalues $\tilde{\lambda}_1', \tilde{\lambda}_2', \ldots, \tilde{\lambda}_n'$ into groups, namely $\mathcal{J}_1 = \{\tilde{\lambda}_{J_0+1}', \ldots, \tilde{\lambda}_{J_1}'\}$, $\mathcal{J}_2 = \{\tilde{\lambda}_{J_1+1}', \ldots, \tilde{\lambda}_{J_2}'\}, \ldots, \mathcal{J}_L = \{\tilde{\lambda}_{J_{L-1}+1}', \ldots, \tilde{\lambda}_{J_L}'\}$, with $J_0 = 0$ and $J_L = n$. We ask that $\tilde{\lambda}_{k+1}' - \tilde{\lambda}_k' > \delta$ for all $k = J_0, J_1, \ldots, J_L$, and $\tilde{\lambda}_{k+1}' - \tilde{\lambda}_k' \leqslant \delta$ for all other $k$. We also denote by $\mathcal{J}(\tilde{\lambda}_i')$ the group where $\tilde{\lambda}_i'$ belong.

The consequence of such splitting is that for any eigenvalue $\tilde{\lambda}'_i$ of $\boldsymbol{L}'$, all $\tilde{\lambda}'_j$ satisfying $\rho(|\tilde{\lambda}'_j - \tilde{\lambda}'_i|) \neq 0$ belong to $\mathcal{J}(\tilde{\lambda}'_i)$. Therefore, we actually have

$$
\min_{\boldsymbol{Q}_i \in O(n)} \left\| \operatorname{concat}\left[ \boldsymbol{v}_j \rho(|\tilde{\lambda}'_j - \tilde{\lambda}'_i|) \right]_{j=1}^n - \operatorname{concat}\left[ \boldsymbol{P}\boldsymbol{v}'_j \rho(|\tilde{\lambda}'_j - \tilde{\lambda}'_i|) \right]_{j=1}^n \boldsymbol{Q}_i \right\|_{\mathrm{F}}
$$

$$
= \min_{\boldsymbol{Q}_i \in O(|\mathcal{J}(\tilde{\lambda}'_i)|)} \left\| \operatorname{concat}\left[ \boldsymbol{v}_j \rho(|\tilde{\lambda}'_j - \tilde{\lambda}'_i|) \right]_{\tilde{\lambda}'_j \in \mathcal{J}(\tilde{\lambda}'_i)} - \operatorname{concat}\left[ \boldsymbol{P}\boldsymbol{v}'_j \rho(|\tilde{\lambda}'_j - \tilde{\lambda}'_i|) \right]_{\tilde{\lambda}'_j \in \mathcal{J}(\tilde{\lambda}'_i)} \boldsymbol{Q}_i \right\|_{\mathrm{F}}.
$$
(45)

Now, for any $\boldsymbol{Q}_i \in O(|\mathcal{J}(\tilde{\lambda}'_i)|)$, we have

$$
\left\| \operatorname{concat}\left[ \boldsymbol{v}_j \rho(|\tilde{\lambda}'_j - \tilde{\lambda}'_i|) \right]_{\tilde{\lambda}'_j \in \mathcal{J}(\tilde{\lambda}'_i)} - \operatorname{concat}\left[ \boldsymbol{P}\boldsymbol{v}'_j \rho(|\tilde{\lambda}'_j - \tilde{\lambda}'_i|) \right]_{\tilde{\lambda}'_j \in \mathcal{J}(\tilde{\lambda}'_i)} \boldsymbol{Q}_i \right\|_{\mathrm{F}}
$$

$$
= \left\| \operatorname{concat}\left[ \boldsymbol{v}_j \rho(|\tilde{\lambda}'_j - \tilde{\lambda}'_i|) - \sum_{k:\tilde{\lambda}'_k \in \mathcal{J}(\tilde{\lambda}'_i)} \boldsymbol{P}\boldsymbol{v}'_k \rho(|\tilde{\lambda}'_k - \tilde{\lambda}'_i|)(\boldsymbol{Q}_i)_{kj} \right]_{\tilde{\lambda}'_j \in \mathcal{J}(\tilde{\lambda}'_i)} \right\|_{\mathrm{F}}
$$
(46)

$$
\leqslant \left\| \operatorname{concat}\left[ \sum_{k:\tilde{\lambda}'_k \in \mathcal{J}(\tilde{\lambda}'_i)} \boldsymbol{P}\boldsymbol{v}'_k \left[ \rho(|\tilde{\lambda}'_j - \tilde{\lambda}'_i|) - \rho(|\tilde{\lambda}'_k - \tilde{\lambda}'_i|) \right](\boldsymbol{Q}_i)_{kj} \right]_{\tilde{\lambda}'_j \in \mathcal{J}(\tilde{\lambda}'_i)} \right\|_{\mathrm{F}}
$$

$$
+ \left\| \operatorname{concat}\left[ \rho(|\tilde{\lambda}'_j - \tilde{\lambda}'_i|) \left( \boldsymbol{v}_j - \sum_{k:\tilde{\lambda}'_k \in \mathcal{J}(\tilde{\lambda}'_i)} \boldsymbol{P}\boldsymbol{v}'_k (\boldsymbol{Q}_i)_{kj} \right) \right]_{\tilde{\lambda}'_j \in \mathcal{J}(\tilde{\lambda}'_i)} \right\|_{\mathrm{F}}
$$
(47)

$$
\leqslant \sum_{j:\tilde{\lambda}'_j \in \mathcal{J}(\tilde{\lambda}'_i)} \left\| \sum_{k:\tilde{\lambda}'_k \in \mathcal{J}(\tilde{\lambda}'_i)} \boldsymbol{P}\boldsymbol{v}'_k \left[ \rho(|\tilde{\lambda}'_j - \tilde{\lambda}'_i|) - \rho(|\tilde{\lambda}'_k - \tilde{\lambda}'_i|) \right](\boldsymbol{Q}_i)_{kj} \right\|
$$

$$
+ \left\| \operatorname{concat}\left[ \rho(|\tilde{\lambda}'_j - \tilde{\lambda}'_i|) \left( \boldsymbol{v}_j - \sum_{k:\tilde{\lambda}'_k \in \mathcal{J}(\tilde{\lambda}'_i)} \boldsymbol{P}\boldsymbol{v}'_k (\boldsymbol{Q}_i)_{kj} \right) \right]_{\tilde{\lambda}'_j \in \mathcal{J}(\tilde{\lambda}'_i)} \right\|_{\mathrm{F}}.
$$
(48)

Now we analyze the two terms in (48). For the first term,

$$
\left\| \sum_{k:\tilde{\lambda}'_k \in \mathcal{J}(\tilde{\lambda}'_i)} \boldsymbol{P}\boldsymbol{v}'_k \left[ \rho(|\tilde{\lambda}'_j - \tilde{\lambda}'_i|) - \rho(|\tilde{\lambda}'_k - \tilde{\lambda}'_i|) \right](\boldsymbol{Q}_i)_{kj} \right\|
$$

$$
= \left\| \operatorname{concat}\left\{ \boldsymbol{P}\boldsymbol{v}'_k \left[ \rho(|\tilde{\lambda}'_j - \tilde{\lambda}'_i|) - \rho(|\tilde{\lambda}'_k - \tilde{\lambda}'_i|) \right] \right\}_{\tilde{\lambda}'_k \in \mathcal{J}(\tilde{\lambda}'_i)} (\boldsymbol{Q}_i)_{\cdot j} \right\|_{\mathrm{F}}
$$
(49)

$$
\leqslant \left\| \operatorname{concat}\left\{ \boldsymbol{P}\boldsymbol{v}'_k \left[ \rho(|\tilde{\lambda}'_j - \tilde{\lambda}'_i|) - \rho(|\tilde{\lambda}'_k - \tilde{\lambda}'_i|) \right] \right\}_{\tilde{\lambda}'_k \in \mathcal{J}(\tilde{\lambda}'_i)} \right\|_{\mathrm{F}} \| (\boldsymbol{Q}_i)_{\cdot j} \|_2
$$
(50)

$$
= \left\| \operatorname{concat}\left\{ \boldsymbol{P}\boldsymbol{v}'_k \left[ \rho(|\tilde{\lambda}'_j - \tilde{\lambda}'_i|) - \rho(|\tilde{\lambda}'_k - \tilde{\lambda}'_i|) \right] \right\}_{\tilde{\lambda}'_k \in \mathcal{J}(\tilde{\lambda}'_i)} \right\|_{\mathrm{F}}
$$
(51)

$$
\leqslant \sum_{k:\tilde{\lambda}'_k \in \mathcal{J}(\tilde{\lambda}'_i)} \| \boldsymbol{P}\boldsymbol{v}'_k \| \left| \rho(|\tilde{\lambda}'_j - \tilde{\lambda}'_i|) - \rho(|\tilde{\lambda}'_k - \tilde{\lambda}'_i|) \right|
$$
(52)

$$
= \sum_{k:\tilde{\lambda}'_k \in \mathcal{J}(\tilde{\lambda}'_i)} \left| \rho(|\tilde{\lambda}'_j - \tilde{\lambda}'_i|) - \rho(|\tilde{\lambda}'_k - \tilde{\lambda}'_i|) \right|.
$$
(53)

Here, (49) translates the first term of (48) into the form of matrix multiplication. Then (50) makes use of Lemma B.3, and (51) further uses the fact that $\boldsymbol{Q}_i$ is orthogonal. Finally, (53) stems from the fact

that $\boldsymbol{v}_k'$ is a normalized eigenvector. Regarding (53), we may discuss two cases. If both $|\tilde{\lambda}_j' - \tilde{\lambda}_i'| \leqslant \delta$ and $|\tilde{\lambda}_k' - \tilde{\lambda}_i'| \leqslant \delta$, then

$$\sum_{k:\tilde{\lambda}_k' \in \mathcal{J}(\tilde{\lambda}_i')} \left| \rho(|\tilde{\lambda}_j' - \tilde{\lambda}_i'|) - \rho(|\tilde{\lambda}_k' - \tilde{\lambda}_i'|) \right|$$

$$\leqslant J_\rho \sum_{k:\tilde{\lambda}_k' \in \mathcal{J}(\tilde{\lambda}_i')} \left| |\tilde{\lambda}_j' - \tilde{\lambda}_i'| - |\tilde{\lambda}_k' - \tilde{\lambda}_i'| \right| \tag{54}$$

$$\leqslant 2\delta J_\rho |\mathcal{J}(\tilde{\lambda}_i')|. \tag{55}$$

If at least one of $|\tilde{\lambda}_j' - \tilde{\lambda}_i'|$ and $|\tilde{\lambda}_k' - \tilde{\lambda}_i'|$ exceeds $\delta$, we may assume without loss of generality that $|\tilde{\lambda}_j' - \tilde{\lambda}_i'| > \delta$. Then $\rho(|\tilde{\lambda}_j' - \tilde{\lambda}_i'|) = \rho(\delta) = 0$ by continuity of $\rho$, and we still have

$$\sum_{k:\tilde{\lambda}_k' \in \mathcal{J}(\tilde{\lambda}_i')} \left| \rho(|\tilde{\lambda}_j' - \tilde{\lambda}_i'|) - \rho(|\tilde{\lambda}_k' - \tilde{\lambda}_i'|) \right|$$

$$= \sum_{k:\tilde{\lambda}_k' \in \mathcal{J}(\tilde{\lambda}_i')} \left| \rho(\delta) - \rho(|\tilde{\lambda}_k' - \tilde{\lambda}_i'|) \right| \tag{56}$$

$$\leqslant J_\rho \sum_{k:\tilde{\lambda}_k' \in \mathcal{J}(\tilde{\lambda}_i')} \left| \delta - |\tilde{\lambda}_k' - \tilde{\lambda}_i'| \right| \tag{57}$$

$$\leqslant 2\delta J_\rho |\mathcal{J}(\tilde{\lambda}_i')|. \tag{58}$$

Therefore, we conclude that

$$\left\| \sum_{k:\tilde{\lambda}_k' \in \mathcal{J}(\tilde{\lambda}_i')} \boldsymbol{P}\boldsymbol{v}_k' \left[ \rho(|\tilde{\lambda}_j' - \tilde{\lambda}_i'|) - \rho(|\tilde{\lambda}_k' - \tilde{\lambda}_i'|) \right] (\boldsymbol{Q}_i)_{kj} \right\| \leqslant 2\delta J_\rho |\mathcal{J}(\tilde{\lambda}_i')|, \tag{59}$$

or

$$\sum_{j:\tilde{\lambda}_j' \in \mathcal{J}(\tilde{\lambda}_i')} \left\| \sum_{k:\tilde{\lambda}_k' \in \mathcal{J}(\tilde{\lambda}_i')} \boldsymbol{P}\boldsymbol{v}_k' \left[ \rho(|\tilde{\lambda}_j' - \tilde{\lambda}_i'|) - \rho(|\tilde{\lambda}_k' - \tilde{\lambda}_i'|) \right] (\boldsymbol{Q}_i)_{kj} \right\| \leqslant 2\delta J_\rho |\mathcal{J}(\tilde{\lambda}_i')|^2. \tag{60}$$

For the second term of (48), we have

$$\left\| \text{concat} \left[ \rho(|\tilde{\lambda}_j' - \tilde{\lambda}_i'|) \left( \boldsymbol{v}_j - \sum_{k:\tilde{\lambda}_k' \in \mathcal{J}(\tilde{\lambda}_i')} \boldsymbol{P}\boldsymbol{v}_k'(\boldsymbol{Q}_i)_{kj} \right) \right]_{\tilde{\lambda}_j' \in \mathcal{J}(\tilde{\lambda}_i')} \right\|_{\text{F}}$$

$$\leqslant \left\| \text{concat} \left[ \boldsymbol{v}_j - \sum_{k:\tilde{\lambda}_k' \in \mathcal{J}(\tilde{\lambda}_i')} \boldsymbol{P}\boldsymbol{v}_k'(\boldsymbol{Q}_i)_{kj} \right]_{\tilde{\lambda}_j' \in \mathcal{J}(\tilde{\lambda}_i')} \right\|_{\text{F}} \tag{61}$$

$$= \left\| \text{concat} \left[ \boldsymbol{v}_j \right]_{\tilde{\lambda}_j' \in \mathcal{J}(\tilde{\lambda}_i')} - \text{concat} \left[ \boldsymbol{P}\boldsymbol{v}_j' \right]_{\tilde{\lambda}_j' \in \mathcal{J}(\tilde{\lambda}_i')} \boldsymbol{Q}_i \right\|_{\text{F}}. \tag{62}$$

Here, (61) uses the fact that $\rho(|\tilde{\lambda}_j' - \tilde{\lambda}_i'|) \in [0, 1]$, and (62) rewrites (61) into matrix multiplication. We further transform (62) into

$$\left\| \text{concat} \left[ \boldsymbol{v}_j \right]_{\tilde{\lambda}_j' \in \mathcal{J}(\tilde{\lambda}_i')} - \text{concat} \left[ \boldsymbol{P}\boldsymbol{v}_j' \right]_{\tilde{\lambda}_j' \in \mathcal{J}(\tilde{\lambda}_i')} \boldsymbol{Q}_i \right\|_{\text{F}}$$

$$\leqslant \left\| \text{concat} \left[ \boldsymbol{v}_j \right]_{\tilde{\lambda}_j' \in \mathcal{J}(\tilde{\lambda}_i')} \boldsymbol{Q}_i^T - \text{concat} \left[ \boldsymbol{P}\boldsymbol{v}_j' \right]_{\tilde{\lambda}_j' \in \mathcal{J}(\tilde{\lambda}_i')} \right\|_{\text{F}} \|\boldsymbol{Q}_i^T\|_2 \tag{63}$$

$$= \left\| \text{concat} \left[ \boldsymbol{P}\boldsymbol{v}_j' \right]_{\tilde{\lambda}_j' \in \mathcal{J}(\tilde{\lambda}_i')} - \text{concat} \left[ \boldsymbol{v}_j \right]_{\tilde{\lambda}_j' \in \mathcal{J}(\tilde{\lambda}_i')} \boldsymbol{Q}_i^T \right\|_{\text{F}}, \tag{64}$$

in which (63) makes use of Lemma B.3, and (64) uses the fact that the spectral norm of an orthogonal matrix is always 1. Now, we may apply Lemma B.1 on (64) to find that there exists $\boldsymbol{Q}_i \in O(|\mathcal{J}(\tilde{\lambda}'_i)|)$, such that

$$\left\| \text{concat} \left[ \boldsymbol{P}\boldsymbol{v}'_j \right]_{\tilde{\lambda}'_j \in \mathcal{J}(\tilde{\lambda}'_i)} - \text{concat} \left[ \boldsymbol{v}_j \right]_{\tilde{\lambda}'_j \in \mathcal{J}(\tilde{\lambda}'_i)} \boldsymbol{Q}_i^T \right\|_{\text{F}}$$

$$\leqslant \frac{\sqrt{8}}{\delta} \min \left\{ \sqrt{|\mathcal{J}(\tilde{\lambda}'_i)|} \cdot \|\boldsymbol{P}\boldsymbol{L}'\boldsymbol{P}^T - \boldsymbol{L}\|_2, \|\boldsymbol{P}\boldsymbol{L}'\boldsymbol{P}^T - \boldsymbol{L}\|_{\text{F}} \right\}. \tag{65}$$

To arrive at (65), we exploit the fact that at boundaries of $\mathcal{J}(\tilde{\lambda}'_i)$ (assumed to be $\tilde{\lambda}'_{J_{\ell-1}+1}$ and $\tilde{\lambda}'_{J_\ell}$), we always have $\tilde{\lambda}'_{J_{\ell-1}+1} - \tilde{\lambda}'_{J_{\ell-1}} > \delta$ and $\tilde{\lambda}'_{J_\ell+1} - \tilde{\lambda}'_{J_\ell} > \delta$. Thus, we end up finding that

$$\left\| \text{concat} \left[ \rho(|\tilde{\lambda}'_j - \tilde{\lambda}'_i|) \left( \boldsymbol{v}_j - \sum_{k:\tilde{\lambda}'_k \in \mathcal{J}(\tilde{\lambda}'_i)} \boldsymbol{P}\boldsymbol{v}'_k (\boldsymbol{Q}_i)_{kj} \right) \right]_{\tilde{\lambda}'_j \in \mathcal{J}(\tilde{\lambda}'_i)} \right\|_{\text{F}}$$

$$\leqslant \frac{\sqrt{8}}{\delta} \min \left\{ \sqrt{|\mathcal{J}(\tilde{\lambda}'_i)|} \cdot \|\boldsymbol{L} - \boldsymbol{P}\boldsymbol{L}'\boldsymbol{P}^T\|_2, \|\boldsymbol{L} - \boldsymbol{P}\boldsymbol{L}'\boldsymbol{P}^T\|_{\text{F}} \right\}. \tag{66}$$

Plugging equations (60) and (66) into (48), we find that $\exists \boldsymbol{Q}_i \in O(|\mathcal{J}(\tilde{\lambda}'_i)|)$, such that

$$\left\| \text{concat} \left[ \boldsymbol{v}_j \rho(|\tilde{\lambda}'_j - \tilde{\lambda}'_i|) \right]_{\tilde{\lambda}'_j \in \mathcal{J}(\tilde{\lambda}'_i)} - \text{concat} \left[ \boldsymbol{P}\boldsymbol{v}'_j \rho(|\tilde{\lambda}'_j - \tilde{\lambda}'_i|) \right]_{\tilde{\lambda}'_j \in \mathcal{J}(\tilde{\lambda}'_i)} \boldsymbol{Q}_i \right\|_{\text{F}}$$

$$\leqslant 2\delta J_\rho |\mathcal{J}(\tilde{\lambda}'_i)|^2 + \frac{\sqrt{8}}{\delta} \min \left\{ \sqrt{|\mathcal{J}(\tilde{\lambda}'_i)|} \cdot \|\boldsymbol{L} - \boldsymbol{P}\boldsymbol{L}'\boldsymbol{P}^T\|_2, \|\boldsymbol{L} - \boldsymbol{P}\boldsymbol{L}'\boldsymbol{P}^T\|_{\text{F}} \right\} \tag{67}$$

$$\leqslant 2n^2 \delta J_\rho + \frac{\sqrt{8}}{\delta} \|\boldsymbol{L} - \boldsymbol{P}\boldsymbol{L}'\boldsymbol{P}^T\|_{\text{F}}. \tag{68}$$

Therefore,

$$\min_{\boldsymbol{Q}_i \in O(n)} \left\| \text{concat} \left[ \boldsymbol{v}_j \rho(|\tilde{\lambda}'_j - \tilde{\lambda}'_i|) \right]_{j=1}^n - \text{concat} \left[ \boldsymbol{P}\boldsymbol{v}'_j \rho(|\tilde{\lambda}'_j - \tilde{\lambda}'_i|) \right]_{j=1}^n \boldsymbol{Q}_i \right\|_{\text{F}}$$

$$\leqslant 2n^2 \delta J_\rho + \frac{\sqrt{8}}{\delta} \|\boldsymbol{L} - \boldsymbol{P}\boldsymbol{L}'\boldsymbol{P}^T\|_{\text{F}}. \tag{69}$$

Plugging (44) and (69) into (40), we get

$$\|f(\tilde{\boldsymbol{V}}_i^{\text{smooth}}) - f(\boldsymbol{P}\tilde{\boldsymbol{V}}_i'^{\text{smooth}})\| \leqslant J_f \left( 2nJ_\rho \|\boldsymbol{L} - \boldsymbol{P}\boldsymbol{L}'\boldsymbol{P}^T\|_2 + 2n^2 \delta J_\rho + \frac{\sqrt{8}}{\delta} \|\boldsymbol{L} - \boldsymbol{P}\boldsymbol{L}'\boldsymbol{P}^T\|_{\text{F}} \right). \tag{70}$$

Combining everything together, we eventually arrive at

$$\|Z(G) - \boldsymbol{P}Z(G')\|_{\text{F}} \leqslant nJ_\psi J_\phi \Bigg[ (\sqrt{n} + 2nJ_\rho J_f) \|\boldsymbol{L} - \boldsymbol{P}\boldsymbol{L}'\boldsymbol{P}^T\|_2$$

$$+ J_f \left( 2n^2 \delta J_\rho + \frac{\sqrt{8}}{\delta} \|\boldsymbol{L} - \boldsymbol{P}\boldsymbol{L}'\boldsymbol{P}^T\|_{\text{F}} \right) \Bigg]. \tag{71}$$

By choosing a $\delta$ value that minimizes the RHS of equation (71), we get

$$\|Z(G) - \boldsymbol{P}Z(G')\|_{\text{F}} \leqslant nJ_\psi J_\phi \big[ (\sqrt{n} + 2nJ_\rho J_f) \|\boldsymbol{L} - \boldsymbol{P}\boldsymbol{L}'\boldsymbol{P}^T\|_2$$

$$+ 4\sqrt[4]{2} J_f \sqrt{J_\rho} n \|\boldsymbol{L} - \boldsymbol{P}\boldsymbol{L}'\boldsymbol{P}^T\|_{\text{F}}^{1/2} \big], \tag{72}$$

which is our desired final result. $\qquad\square$

## C  OTHER RELATED WORKS

**Expressive GNNs.**  As is shown by Xu et al. (2018), the expressive power of MPNNs is upper-bounded by that of 1-dimensional Weisfeiler-Leman test (1-WL). This implies that MPNNs can fail to discriminate many non-isomorphic graph pairs, potentially leading to their weakness in capturing important structural information or multi-node interactions. A great number of works have attempted to improve the expressive power of GNNs, in the sense that to make them better either at solving the graph isomorphism problem (GI), or at approximating certain graph functions. Those existing works can be roughly categorized into three families: (1) methods utilizing additional combinatorial features (Barceló et al., 2021; Bouritsas et al., 2022; Li et al., 2020); (2) methods applying message passing among higher-order tuples of nodes, or *higher-order GNNs* (Bodnar et al., 2021; Feng et al., 2023; Maron et al., 2018; 2019; Morris et al., 2019; 2020; Zhang et al., 2023; Zhou et al., 2023b;c); (3) methods decomposing input graphs into bags of subgraphs, or *subgraph GNNs* (Bevilacqua et al., 2024; Cotta et al., 2021; Frasca et al., 2022; Huang et al., 2023; Kong et al., 2023; Qian et al., 2022; You et al., 2021; Zhang & Li, 2021; Zhou et al., 2023b). While methods belonging to class (1) enjoy the lowest complexities, they often generalize worse due to their use of hand-crafted features. On the contrary, higher-order GNNs and subgraph GNNs bring more systematic gains to the expressive power, but their computational complexities are much higher than MPNNs. Hence, a trade-off between expressive power and efficiency is an important issue for the design of expressive GNNs.

**Graph transformers.**  Graph transformers (Chen et al., 2022; Dwivedi et al., 2021; Rampášek et al., 2022; Wang et al., 2024; Ying et al., 2021b) treat each node within a graph as a separate token, and use a standard transformer architecture to update node features (or embeddings of tokens). With attention mechanism, graph transformers take into account the interactions between all pairs of nodes (instead of only connected node pairs, as in traditional MPNNs), and are naturally good at capturing long-range interactions (Dwivedi et al., 2022). One of the central issues regarding graph transformers is the design of structural and positional encodings of nodes, in order to make transformers aware of adjacency information. Kim et al. (2022); Zhou et al. (2024) analyze the theoretical expressive power of graph transformers and their high-order versions as well as the effects of positional encodings.

## D  EXPERIMENTAL DETAILS

### D.1  DATASET DESCRIPTIONS

The statistics of used datasets in the paper (except for DrugOOD) are summarized in Table 5.

Table 5: Overview of the datasets used in the paper.

| Dataset | #Graphs | Avg. # nodes | Avg. # edges | Prediction level | Prediction task | Metric |
|---|---|---|---|---|---|---|
| QM9 | 130,000 | 18.0 | 37.3 | graph | regression | Mean Abs. Error |
| ZINC | 12,000 | 23.2 | 24.9 | graph | regression | Mean Abs. Error |
| Alchemy | 202,579 | 10.0 | 10.4 | graph | regression | Mean Abs. Error |
| PCQM-Contact | 529,434 | 30.1 | 61.0 | inductive link | link ranking | MRR |
| CLUSTER | 12,000 | 117.20 | 4,301.72 | node | classification | Accuracy |
| PATTERN | 14,000 | 117.47 | 4,749.15 | node | classification | Accuracy |
| ogbg-molhiv | 41,127 | 25.5 | 27.5 | graph | classification | AUROC |

### D.2  IMPLEMENTATION DETAILS

#### D.2.1  ARCHITECTURE DESIGN

To implement OGE-Aug practically, the central issue is to choose a proper orthogonal-group invariant encoder $f$ in equation (7). In our experiments, we uniformly adopt a point cloud network architecture similar to the one proposed in (Finkelshtein et al., 2022). We provide the detailed implementation in Algorithm 1. Here, $\text{Linear}^{Q,b}_{\text{shape}_1 \to \text{shape}_2}$ or $\text{Linear}^{Q}_{\text{shape}_1 \to \text{shape}_2}$ means a linear transformation operating on the last dimension of $\text{shape}_1$ and transforming it into $\text{shape}_2$, either with or without bias $b$. In

---

**Algorithm 1:** Practical implementation of OGE-Aug.

---

**Data:** Node features $\boldsymbol{X} \in \mathbb{R}^{n \times d}$, the matrix of Laplacian eigenvectors $\boldsymbol{V} = (\boldsymbol{v}_1, \ldots, \boldsymbol{v}_n) \in \mathbb{R}^{n \times n}$, and $\tilde{\boldsymbol{V}}_1^{\text{smooth}}, \ldots, \tilde{\boldsymbol{V}}_n^{\text{smooth}} \in \mathbb{R}^{n \times n}$ as defined in equation (29).

**Result:** Node feature augmentations $\boldsymbol{Z} \in \mathbb{R}^{n \times h}$.

**(a) Preparation.** Given weight matrices $\boldsymbol{Q}_0^{\text{init}} \in \mathbb{R}^{d \times h}$, $\boldsymbol{b}_0^{\text{init}} \in \mathbb{R}^h$, $\boldsymbol{Q}_1^{\text{init}}, \boldsymbol{Q}_2^{\text{init}} \in \mathbb{R}^{1 \times h}$,

$\mathbf{W}^{(0)} \leftarrow \text{Linear}_{(\cdot,d) \to (\cdot,h)}^{\boldsymbol{Q}_0^{\text{init}}, \boldsymbol{b}_0^{\text{init}}}(\boldsymbol{X})$ ;  # $\mathbf{W}^{(0)} \in \mathbb{R}^{n \times h}$

$\mathbf{W}^{(1)} \leftarrow \text{Linear}_{(\cdot,\cdot,1) \to (\cdot,\cdot,h)}^{\boldsymbol{Q}_1^{\text{init}}}(\boldsymbol{V}.\text{unsqueeze}(-1))$ ;  # $\mathbf{W}^{(1)} \in \mathbb{R}^{n \times n \times h}$

$\mathbf{W}_{a,j,k,:}^{(2)} \leftarrow \text{Linear}_{1 \to h}^{\boldsymbol{Q}_2^{\text{init}}} \left[ (\tilde{\boldsymbol{V}}_j^{\text{smooth}})_{ak}(\tilde{\boldsymbol{V}}_k^{\text{smooth}})_{aj} \right]$ ;  # $\mathbf{W}^{(2)} \in \mathbb{R}^{n \times n \times n \times h}$

**(b) Updates.** Alternately apply the following two types of layers for $N$ times.

    **(i) Tensor product layer.** Given input $\mathbf{W}^{(0)}, \mathbf{W}^{(1)}, \mathbf{W}^{(2)}$, weight matrices $\boldsymbol{Q}_0^{\text{prod}}, \boldsymbol{Q}_1^{\text{prod}}$, $\boldsymbol{Q}_2^{\text{prod}}, \boldsymbol{R}_0^{\text{prod}}, \boldsymbol{R}_1^{\text{prod}}, \boldsymbol{R}_2^{\text{prod}} \in \mathbb{R}^{h \times h}$, $\boldsymbol{b}_0^{\text{prod}} \in \mathbb{R}^h$ and $\boldsymbol{c} \in \mathbb{R}^{3 \times 3}$,

      ① $\mathbf{W}_{\text{norm}}^{(1)} \leftarrow \text{Normalize}(\mathbf{W}^{(1)}, \dim = 1)$;

      ② $\mathbf{W}_{\text{norm}}^{(2)} \leftarrow \text{Normalize}(\mathbf{W}^{(2)}, \dim = (1, 2))$;

      ③ $\tilde{\mathbf{W}}^{(0)}, \tilde{\mathbf{W}}^{(1)}, \tilde{\mathbf{W}}^{(2)} \leftarrow \sigma \left( \text{Linear}_{(\cdot,h) \to (\cdot,h)}^{\boldsymbol{Q}_0^{\text{prod}}, \boldsymbol{b}_0^{\text{prod}}}(\mathbf{W}^{(0)}) \right), \text{Linear}_{(\cdot,\cdot,h) \to (\cdot,\cdot,h)}^{\boldsymbol{Q}_1^{\text{prod}}}(\mathbf{W}_{\text{norm}}^{(1)})$,

        $\text{Linear}_{(\cdot,\cdot,h) \to (\cdot,\cdot,h)}^{\boldsymbol{Q}_2^{\text{prod}}}(\mathbf{W}_{\text{norm}}^{(2)})$, where $\sigma(\cdot)$ is a normalization layer followed by element-wise SiLU;

      ④ $\mathbf{W}_{ij}^{(0)} \leftarrow \mathbf{W}_{ij}^{(0)} + \text{matmul}\Big[ c_{00}\mathbf{W}_{ij}^{(0)}\tilde{\mathbf{W}}_{ij}^{(0)} + c_{01}\sum_k \mathbf{W}_{ikj}^{(1)}\tilde{\mathbf{W}}_{ikj}^{(1)} +$

        $c_{02}\sum_{k,\ell} \mathbf{W}_{ik\ell j}^{(2)}\tilde{\mathbf{W}}_{ik\ell j}^{(2)}, \boldsymbol{R}_0^{\text{prod}} \Big]$;

      ⑤ $\mathbf{W}_{ikj}^{(1)} \leftarrow \mathbf{W}_{ikj}^{(1)} + \text{matmul}\Big[ c_{10}\mathbf{W}_{ikj}^{(1)}\tilde{\mathbf{W}}_{ij}^{(0)} + c_{12}\sum_\ell \mathbf{W}_{i\ell j}^{(1)}\tilde{\mathbf{W}}_{ik\ell j}^{(2)}, \boldsymbol{R}_1^{\text{prod}} \Big]$;

      ⑥ $\mathbf{W}_{ik\ell j}^{(2)} \leftarrow \mathbf{W}_{ik\ell j}^{(2)} + \text{matmul}\Big[ c_{20}\mathbf{W}_{ik\ell j}^{(2)}\tilde{\mathbf{W}}_{ij}^{(0)} + c_{11}\rho^2(|\tilde{\lambda}_k - \tilde{\lambda}_\ell|)\mathbf{W}_{ikj}^{(1)}\tilde{\mathbf{W}}_{i\ell j}^{(1)} +$

        $c_{22}\rho^2(|\tilde{\lambda}_k - \tilde{\lambda}_\ell|)\sum_m \mathbf{W}_{ikmj}^{(2)}\tilde{\mathbf{W}}_{im\ell j}^{(2)}, \boldsymbol{R}_2^{\text{prod}} \Big]$;

    **(ii) Message passing layer.** Given input $\mathbf{W}^{(0)}, \mathbf{W}^{(1)}, \mathbf{W}^{(2)}$, adjacency matrix $\boldsymbol{A}$ and weight matrices $\boldsymbol{Q}_0^{\text{msg}}, \boldsymbol{Q}_1^{\text{msg}}, \boldsymbol{Q}_2^{\text{msg}} \in \mathbb{R}^{h \times h}$, $\boldsymbol{b}_0^{\text{msg}} \in \mathbb{R}^h$,

      ① $\mathbf{W}_{\text{norm}}^{(1)} \leftarrow \text{Normalize}(\mathbf{W}^{(1)}, \dim = 1)$;

      ② $\mathbf{W}_{\text{norm}}^{(2)} \leftarrow \text{Normalize}(\mathbf{W}^{(2)}, \dim = (1, 2))$;

      ③ $\tilde{\mathbf{W}}^{(0)}, \tilde{\mathbf{W}}^{(1)}, \tilde{\mathbf{W}}^{(2)} \leftarrow \sigma \left( \text{Linear}_{(\cdot,h) \to (\cdot,h)}^{\boldsymbol{Q}_0^{\text{msg}}, \boldsymbol{b}_0^{\text{msg}}}(\mathbf{W}^{(0)}) \right), \text{Linear}_{(\cdot,\cdot,h) \to (\cdot,\cdot,h)}^{\boldsymbol{Q}_1^{\text{msg}}}(\mathbf{W}_{\text{norm}}^{(1)})$,

        $\text{Linear}_{(\cdot,\cdot,h) \to (\cdot,\cdot,h)}^{\boldsymbol{Q}_2^{\text{msg}}}(\mathbf{W}_{\text{norm}}^{(2)})$, where $\sigma(\cdot)$ is a normalization layer followed by element-wise SiLU;

      ④ $\mathbf{W}_{i:}^{(0)} \leftarrow \mathbf{W}_{i:}^{(0)} + \sum_k A_{ik}\tilde{\mathbf{W}}_{k:}^{(0)}$;

      ⑤ $\mathbf{W}_{i::}^{(1)} \leftarrow \mathbf{W}_{i::}^{(1)} + \sum_k A_{ik}\tilde{\mathbf{W}}_{k::}^{(1)}$;

      ⑥ $\mathbf{W}_{i:::}^{(2)} \leftarrow \mathbf{W}_{i:::}^{(2)} + \sum_k A_{ik}\tilde{\mathbf{W}}_{k:::}^{(2)}$;

**(c) Output.** $\boldsymbol{Z} \leftarrow \mathbf{W}^{(0)}$.

---

PyTorch, such operations would translate to `nn.Linear` modules. The operator `matmul` operates similarly to `torch.matmul`. The function $\rho(x)$ takes the form

$$\rho(x) = \begin{cases} \frac{1}{2}\left(1 + \cos\frac{\pi x}{\delta}\right), & 0 \leqslant x \leqslant \delta, \\ 0, & x > \delta, \end{cases} \tag{73}$$

where $\delta$ is a hyperparameter.

We now discuss the complexity of Algorithm 1 as well as its connections to our theoretically proposed OGE-Aug (Definition 4.1). It is not hard to notice that the most computationally costly steps of Algorithm 1 are those to compute $\rho^2(|\tilde{\lambda}_k - \tilde{\lambda}_\ell|)\sum_m \mathbf{W}_{ikmj}^{(2)}\tilde{\mathbf{W}}_{im\ell j}^{(2)}$ and $\sum_k A_{ik}\tilde{\mathbf{W}}_{k:::}^{(2)}$. If we use dense matrices to store all the necessary data, the time complexity to compute those two terms are $O(n^4)$ and $O(n^2 m)$, where $n$ and $m$ refer to the number of nodes and edges of $G$, respectively.

Nevertheless, since the smoothing function $\rho(\cdot)$ is only non-zero when its argument is sufficiently close to zero, we find that $\mathbf{W}^{(2)}_{i::j}$ is a sparse matrix with only $O(n \max_j \mu_j)$ non-zero elements, for each $i = 1, \ldots, n$ and $j = 1, \ldots, h$. Here, $\max_j \mu_j$ means the maximum multiplicity of $G$'s Laplacian eigenvalues. Therefore, by storing $\mathbf{W}^{(2)}$ as a sparse matrix, the above two terms can be computed in $O(n^2 \max_j \mu_j^2)$ and $O(m \max_j \mu_j)$ time respectively, resulting a practical time complexity of $O((n^2 \max_j \mu_j + m) \cdot \max_j \mu_j)$, which is generally lower than $O(n^3)$.

We remark that although Algorithm 1 uses only tensors up to second order, it is not hard to generalize Algorithm 1 to accommodate higher-order tensors based on $\tilde{V}_1^{\text{smooth}}, \ldots, \tilde{V}_n^{\text{smooth}}$, resulting in a model with higher complexities and better expressive power. When the tensor order reaches $n$, our implementation of OGE-Aug can produce universally expressive graph representations, recovering our theoretical result. Since this would entail an unaffordable complexity of $O(n \cdot n^n) = n \exp(\tilde{O}(n))$, Algorithm 1 is adopted practically instead, at the cost of some expressivity.

Finally, we point out that Algorithm 1 does not tightly follow equation (7), in that (i) apart from using $V_1^{\text{smooth}}, \ldots, V_K^{\text{smooth}}$ (to build second-order tensors), Algorithm 1 also uses information directly from the raw Laplacian eigenvectors (to build first-order tensors), and that (ii) Algorithm 1 allows mixing of $V_j^{\text{smooth}}$ with different $j$. Despite those differences, Algorithm 1 maintains the key idea of OGE-Aug: only information from two Laplacian eigenspaces whose corresponding eigenvalues are "not too far away" from each other would be multiplied into $\mathbf{W}^{(2)}$, and the algorithm has no explicit dependence on the multiplicities of Laplacian eigenvalues. Thus, the stability result demonstrated in Proposition 4.3 can similarly hold for Algorithm 1, though the accurate bound may be different.

### D.2.2 OTHER DETAILS OF THE PRACTICAL IMPLEMENTATION

We implement OGE-Aug with the PyGHO library (Wang & Zhang, 2023). To integrate OGE-Aug with other base models including MPNN and graph transformers, we also implement our methods building on the GraphGPS (Rampášek et al., 2022) code base, where we build OGE-Aug as a plug-and-play module. The module takes in Laplacians as inputs and processes the eigenvalues/eigenvectors using *# PE layers* with dimension *PE hidden dim*, and outputs an embedding with dimension *PE dim*; see Table 6 for detailed settings. In this module, we use permutation-equivariant set function (Zaheer et al., 2017) to process the eigenvalues and multiply the eigenvalue embeddings to the eigenvectors. Moreover, we also multiply eigenvectors with eigenvectors to initialize the higher order representations. After that, this module will product each node's representation with its neighbors' and update the representation iteratively. The embedding is then combined with other node features and other optional positional encodings, then fed jointly into downstream layers (which consist of various GNN and graph transformer modules). Therefore, OGE-Aug can be either used solely or integrated easily with arbitrary backbones.

We also implement a version where OGE-Aug modules act on the embeddings of nodes and edges, which can be viewed as operating on weighted or latent Laplacians incorporating node and edge features. However, we experimentally find that processing the original Laplacians with OGE-Aug and encoding the node/edge features separately via other encoders (as explained above) yields better performance.

In addition, to make OGE-Aug more robust, we add a small-scale noise (typically a Gaussian noise with mean zero and variance $10^{-5}$) to the Laplacians in the training process. We also randomly permute the Laplacians and do inverse permutation to the output eigenvectors to simulate the noise caused by the permutation and the numerical algorithm. We use the original Laplacians in the inference stage.

### D.3 EXPERIMENTAL SETTINGS

As explained earlier, we integrate our OGE-Aug with the GraphGPS code base, and thus also follow their experimental settings. With only mild hyperparameter search, we achieve SOTA or highly competitive results on all datasets. The adopted hyperparameters in our experiments are summarized in Table 6.

Here $^\dagger$ for QM9 suggests that experiments on these four targets $U_0, U, G, H$ are conducted using the PyGHO code version without GraphGPS. $^*$ for ZINC means that the transformer is not necessary

Table 6: Hyperparameters of the experiments.

| Hyperparameters | QM9 | ZINC | Alchemy | PCQM-Contact |
|---|---|---|---|---|
| # Layers | 10 | 10 | 16 | 6 |
| Hidden dim | 64 | 64 | 128 | 96 |
| MPNN | GINE | GINE | GINE | GatedGCN |
| Attention | Transformer$^{\dagger}$ | Transformer$^{*}$ | - | Transformer |
| # Heads | 4 | 4 | - | 4 |
| Dropout | 0 | 0 | 0 | 0 |
| Attention dropout | 0.2 | 0.5 | - | 0.1 |
| Graph pooling | sum | sum | sum | edge dot |
| Positional encoding | OGE-Aug(29) | OGE-Aug(37) | OGE-Aug(12) | OGE-Aug + LapPE |
| PE hidden dim | 64 | 64 | 64 | 32 |
| PE dim | 28 | 28 | 28 | 16 |
| PE # layer | 4 | 4 | 4 | 3 |
| Batch size | 256 | 32 | 128 | 64 |
| Learning rate | 0.001 | 0.001 | 0.001 | 0.0005 |
| # Epochs | 500 | 2000 | 1000 | 100 |
| # Warmup epochs | 50 | 50 | 50 | 10 |
| Weight decay | 1e-5 | 1e-5 | 1e-5 | 0 |
| # Parameters | 783249 | 617677 | 1968352 | 845632 |
| Time (epoch/total) | 139s/19.3h | 28s/15.6h | 5s/1.4h | 1541s/42.8h |

Table 7: Five-run results on CLUSTER, PATTERN and ogbg-molhiv.

| Method | CLUSTER (Acc ↑) | PATTERN (Acc ↑) | ogbg-molhiv (AUROC ↑) |
|---|---|---|---|
| GCN | $68.50 \pm 0.98$ | $71.89 \pm 0.34$ | $75.99 \pm 1.19$ |
| GIN | $64.72 \pm 1.55$ | $85.39 \pm 0.14$ | $77.07 \pm 1.49$ |
| GAT | $70.59 \pm 0.45$ | $78.27 \pm 0.19$ | - |
| GatedGCN | $73.84 \pm 0.33$ | $85.57 \pm 0.09$ | $78.74 \pm 1.19$ |
| SAN | $76.69 \pm 0.65$ | $86.58 \pm 0.37$ | $77.85 \pm 2.47$ |
| K-Subgraph SAT | $77.86 \pm 0.10$ | $86.85 \pm 0.37$ | - |
| GraphGPS | $78.02 \pm 0.18$ | $86.69 \pm 0.59$ | $78.80 \pm 1.01$ |
| Exphormer | $78.07 \pm 0.04$ | $86.74 \pm 0.15$ | - |
| OGE-Aug | $\mathbf{78.33 \pm 0.13}$ | $\mathbf{86.87 \pm 0.33}$ | $\mathbf{80.01 \pm 0.59}$ |

- actually we can achieve highly competitive results even without global attention. When we use transformers, we reduce the PE hidden dimension to 32, PE dimension to 16, and PE # layers to 3, resulting 505905 total number of parameters and 14.9h total training time, which are both less than the case without transformers.

### D.4   OTHER EXPERIMENTAL RESULTS

**Other graph benchmarks.**   We evaluate the performance of OGE-Aug on three additional graph learning benchmarks: CLUSTER (Dwivedi et al., 2023), PATTERN (Dwivedi et al., 2023) and ogbg-molhiv (Hu et al., 2021). CLUSTER and PATTERN are node classification datasets, while ogbg-molhiv is a graph classification dataset. The results are summarized in Table 7. We quote the baseline results directly from Rampášek et al. (2022) and Shirzad et al. (2023). One may find that OGE-Aug outperforms all baselines on the three datasets.

**OOD benchmarks.**   We evaluate the OOD performance of OGE-Aug on DrugOOD (Ji et al., 2022), an OOD benchmark for drug discovery. We consider three domains on which distribution shifts exist, namely Assay (which assay the molecule belongs to), Scaffold (core structure of the molecule) and Size (size of the molecule). For each domain, the dataset is divided into five splits: the training set, the in-distribution (ID) validation/test sets, and the out-of-distribution (OOD) validation/test sets. The data distribution of OOD splits is different from that of ID splits regarding the specific domain. The

Table 8: AUROC (the larger, the better) results on DrugOOD.

| Domain | Method | ID-Val (AUROC) | ID-Test (AUROC) | OOD-Val (AUROC) | OOD-Test (AUROC) |
|---|---|---|---|---|---|
| Assay | No PE | 92.92 | 92.89 | 71.02 | 71.68 |
| | PEG | 92.51 | 92.57 | 70.86 | 71.98 |
| | SignNet | 92.26 | 92.43 | 70.16 | 72.27 |
| | BasisNet | 88.96 | 89.42 | 71.19 | 71.66 |
| | SPE | 92.84 | **92.94** | 71.26 | 72.53 |
| | OGE-Aug | **94.88** | 86.75 | **82.26** | **73.73** |
| Scaffold | No PE | 96.56 | 87.95 | 79.07 | 68.00 |
| | PEG | 95.65 | 86.20 | 79.17 | 69.15 |
| | SignNet | 95.48 | 86.73 | 77.81 | 66.43 |
| | BasisNet | 85.80 | 78.44 | 73.36 | 66.32 |
| | SPE | **96.32** | **88.12** | **80.03** | **69.64** |
| | OGE-Aug | 95.02 | 86.54 | 78.67 | 65.94 |
| Size | No PE | **93.78** | **93.60** | **82.76** | **66.04** |
| | PEG | 92.46 | 92.67 | 82.12 | 66.01 |
| | SignNet | 93.30 | 93.20 | 80.67 | 64.03 |
| | BasisNet | 86.04 | 85.51 | 75.97 | 60.79 |
| | SPE | 92.46 | 92.67 | 82.12 | 66.02 |
| | OGE-Aug | 94.65 | 84.88 | 78.44 | 64.64 |

task is graph-level binary classification, i.e., to predict whether the drug is active. We use AUROC as the evaluation metric.

The experimental results are shown in Table 8. We choose PE methods from (Huang et al., 2024) as our baselines. Our OGE-Aug outperforms all baselines on the Assay domain, and achieves comparable results on Scaffold and Size domains. Moreover, the performance of our method is better than that of BasisNet on 5 out of the 6 OOD evaluation targets, verifying the benefits of possessing theoretically guaranteed stability.

**Ablation studies.** Finally, we study the effect of the smoothing function $\rho(\cdot)$ in OGE-Aug. We use ZINC as the evaluation dataset. We take GINE as the base model, and apply either Vanilla OGE-Aug, or OGE-Aug with different smoothing functions $\rho(\cdot)$ (all of them taking the form of equation (73) but with different hyperparameters $\delta$). The results are shown in Table 9.

We find that applying Vanilla OGE-Aug instead of OGE-Aug leads to significant performance drop, which verifies the importance of ensuring stability by introducing the smoothing function $\rho$. We also observe that as long as the hyperparameter $\delta$ is not too close to zero, the performance varies little with different choices of $\delta$.

Table 9: Ablation studies on ZINC.

| Method | MAE ($\downarrow$) |
|---|---|
| Vanilla OGE-Aug | 0.098 |
| OGE-Aug ($\delta = 5 \times 10^{-3}$) | 0.066 |
| OGE-Aug ($\delta = 5 \times 10^{-2}$) | 0.066 |
| OGE-Aug ($\delta = 5 \times 10^{-1}$) | 0.065 |

