# OpenReview forum: "Towards Stable, Globally Expressive Graph Representations with Laplacian Eigenvectors"
_ICLR.cc/2025/Conference — Submitted to ICLR 2025_

### Official Review · Reviewer_eRwE · 2024-10-30

**Soundness:** 3
**Presentation:** 2
**Contribution:** 2
**Rating:** 5
**Confidence:** 4

**Summary:**

This paper improves GNNs by incorporating Laplacian eigenvectors to capture global graph properties, addressing limitations in local message-passing approaches.  The proposed method uses learnable O(p)-invariant representations for each Laplacian eigenspace of dimension p and handles close eigenvalues in a smooth fashion, enhancing robustness and expressivity. Experiments show competitive performance, especially in capturing global graph characteristics.

**Strengths:**

1. The proposed OGE-Aug method is theoretically robust, demonstrating strong expressiveness and stability. The underlying motivation is both novel and compelling.
2.  Empirical experiments indicate that OGE-Aug performs competitively and holds promise for real-world applications.
3. The theoretical contributions of this paper are sufficient, and the proofs appear sound.

**Weaknesses:**

1. The proposed OGE-Aug method is intricate, highly complex, and challenging to implement. In this paper, the authors present a simplified version using a Cartesian tensor-based point cloud network, which introduces a gap between the theoretical framework and practical implementation.
2. This paper's writing is rough and difficult to read and follow. First, I recommend that the authors adopt the notation guidelines suggested by the ICLR template. In the current version, symbols such as V are used interchangeably to represent different elements, like sets and matrices, which makes the paper challenging to interpret. Second, the implementation of OGE-Aug should be presented with more detail, ideally including a pseudocode.
3. There is no available code to reproduce the results or gain a deeper understanding of the method’s details.
4. In the experiments, the primary baselines are Laplacian-based embedding methods, but some recent state-of-the-art methods, such as Exphormer [1] on the PCQM-Contact dataset, are missing.
5. While the experiments include four different datasets or benchmarks, incorporating additional datasets or widely-used benchmarks, such as the OGB benchmark (e.g., ogbg-molhiv, ogbg-molpcba), could further strengthen the evaluation.

[1] Hamed Shirzad, Ameya Velingker, Balaji Venkatachalam, Danica J. Sutherland, Ali Kemal Sinop. Proceedings of the 40th International Conference on Machine Learning, PMLR 202:31613-31632, 2023.

**Questions:**

1. Please begin by clarifying the weaknesses section.
2. Could the authors provide an empirical comparison of training costs with the most competitive baselines?

---

> ### Author Response · Authors · 2024-12-01
>
> Thank you for your insightful comments and valuable questions. We respond to all the weaknesses and questions below. As we have revised our paper and given some responses in our public comments titled **"Paper revision"** (shortened as **PR**) and **"Further comments regarding Q1 of Reviewer wrYh and Q1 of Reviewer 9aT4"** (shortened as **FC**), in the following we will directly quote them if our intended response is already there.
>
> **For W1.** We remark that the transition from Vanilla OGE-Aug (Definition 3.6) to OGE-Aug (Definition 4.1) is quite natural, by merely replacing the raw Laplacian eigenvectors with the "smoothed" eigenvectors. Although we admit there is a gap between our proposed OGE-Aug (Definition 4.1) and our implemented version (Algorithm 1), we claim that in the limiting case (by increasing tensor order to $O(n)$), our practically implemented OGE-Aug has the potential to recover the universal expressivity as is the case for Definition 3.6 and 4.1. Finally, our rationale to implement Algorithm 1 (instead of applying a universal point cloud network, as our theory indicates) is mainly due to the unacceptable complexity of $n\\cdot\\exp(\\tilde{O}(n))$ of applying such universal point cloud networks. There is an inevitable trade-off between the practicality of our model and the gap between our theory and implementation.
>
> **For W2.** We have followed the suggestions, adopting notation guidelines suggested by ICLR template. Please refer to the 1st bullet point of **PR** as well as our revised paper. As for the pseudocode of our practically implemented OGE-Aug, please refer to the 4th bullet point of **PR** as well as Algorithm 1 in our revised paper.
>
> **For W3.** We have already provided code to reproduce our experimental results in our supplementary materials.
>
> **For W4.** Please refer to the 6th bullet point of **PR** as well as our revised paper.
>
> **For W5.** Please refer to the 7th bullet point of **PR** as well as our revised paper. We provide the experimental results on the ogbg-molhiv benchmarks. We will update the result on ogbg-molpcba once they are ready.
>
> **For Q2.** In our Table 6 we provide the training costs of our method on different benchmark dataset. We now compare them to some baselines. We take ZINC as our benchmarking dataset. In the following table, **Time** means training time per epoch. The baseline information is quoted from [1].
>
> | | PST|	SUN	|SSWL	|PPGN|	Graphormer	|SAN-GPS	|OGE-Aug (ours)|
> |:---:|:---:|:---:|:---:|:---:|:---:|:---:|:---:|
> |**Time/s**|	15|	21|	45	|20|	124|	79|	28|
>
> One may see that our model runs much faster than Graphormer [2] or SAN [3], even though our model also uses the graph transformer architecture with positional encodings computed from Laplacian eigenvectors. Moreover, our model is faster than subgraph GNNs like SSWL [4]. Those results verify the relatively good efficiency of our method.
>
> [1] Xiyuan Wang, Pan Li, Muhan Zhang. Graph as Point Set. arXiv:2405.02795.
>
> [2] Chengxuan Ying, Tianle Cai, Shengjie Luo, Shuxin Zheng, Guolin Ke, Di He, Yanming Shen, Tie-Yan Liu. Do Transformers Really Perform Bad for Graph Representation? arXiv:2106.05234.
>
> [3] Devin Kreuzer, Dominique Beaini, William L. Hamilton, Vincent Létourneau, Prudencio Tossou. Rethinking Graph Transformers with Spectral Attention. arXiv:2106.03893.
>
> [4] Bohang Zhang, Guhao Feng, Yiheng Du, Di He, Liwei Wang. A Complete Expressiveness Hierarchy for Subgraph GNNs via Subgraph Weisfeiler-Lehman Tests. arXiv:2302.07090.

---

> > ### Comment · Reviewer_eRwE · 2024-12-02
> >
> > Thanks to the authors' responses, they have addressed my concerns. However, the revised paper shows that the results in Tables 1, 2, 3, and 4 (main experiments) do not clearly demonstrate that the proposed method significantly outperforms the previous approach, in other words, no substantial improvement over the previous method.

---

### Official Review · Reviewer_9aT4 · 2024-10-31

**Soundness:** 2
**Presentation:** 3
**Contribution:** 2
**Rating:** 6
**Confidence:** 4

**Summary:**

This paper addresses the problem (established in [2]) of how to incorporate Laplacian eigenvectors into message-passing neural networks (MPNNs) to enhance expressivity while maintaining invariance to the symmetric group. The authors propose two main architectures to achieve this.

First, they present a natural method for integrating equivariant point cloud architectures into existing MPNNs. They then show that when using this method with a universal point cloud architecture, a universal GNN model can be constructed.

The paper then tackles the recently established problem (see [1]) of handling these eigenvectors in a stable manner. They propose a second architecture that, while sacrificing some expressivity compared to the first model, achieves provable stability bounds.

The paper concludes with empirical validation, showing that the second architecture achieves competitive, and in some cases state-of-the-art results across several standard graph benchmarks.

[1]  Yinan Huang, William Lu, Joshua Robinson, Yu Yang, Muhan Zhang, Stefanie Jegelka, and Pan Li. On the stability of expressive
positional encodings for graph neural networks. arXiv preprint arXiv:2310.02579, 2023.

[2] Derek Lim, Joshua Robinson, Lingxiao Zhao, Tess Smidt, Suvrit Sra, Haggai Maron, and Stefanie Jegelka. Sign and basis invariant networks for spectral graph representation learning. arXiv preprint arXiv:2202.13013, 2022.

**Strengths:**

1. The paper offers a simple and straightforward way to incorporate Laplacian eigenvectors into GNNs, making it easy to apply to existing architectures.

2. The theoretical result showing that the first approach can achieve universality with a model complexity of $n \cdot \exp(\text{max}(\mu))$ leverages the sparsity of natural graphs in an interesting way (most graphs have low values of $\mu$).

3. The approach is validated with competitive, and in some cases state-of-the-art, results across several common graph benchmarks, supporting its practical usefulness.

**Weaknesses:**

1. The paper's novelty is somewhat limited. The idea of incorporating Laplacian eigenvectors into MPNNs while respecting the symmetries of the orthogonal group $O(d)$ was already discussed in [2], making the use of point cloud architectures a natural but not particularly novel approach. Similarly, the importance of stability when dealing with Laplacian eigenvectors was pointed out in [1], and the way the second model is derived from the first resembles the approach proposed in [1] as well.

2. The introduction and title of the paper emphasize improved global performance, but this is not explored in depth throughout the rest of the paper. A more convincing theoretical argument explaining why the proposed model enhances "global expressivity" would be beneficial. Additionally, synthetic experiments demonstrating that the model outperforms others in learning global properties of graphs would strengthen the claims. Lastly, a brief recap of the types of global information that can be inferred from Laplacian eigenvalues and eigenvectors would provide useful context and support the discussion.

3. The theory section could be more cohesive. The paper introduces a first model, notes its likely instability, and then replaces it with a second model that is provably stable but less expressive. All experiments are then performed on the second model. However, the discussion regarding expressivity primarily focuses on the first model, leaving the expressivity of the second under explored despite being the one used in practice. Additionally, the authors define stability in one way but then prove the stability of their model using a slightly different notion. This could benefit from clearer alignment and explanation.

4. The paper [3] provides a similar analysis, outlining expressivity bounds of previous methods using laplacian eigenvectors and proposing a new architecture. A comparison between the results of that work and the current paper could offer valuable insights and strengthen the positioning of this paper’s contributions.

5. While the results on the QM9 benchmark are promising, it would be helpful to include a comparison with other methods that also use Laplacian eigenvectors on this dataset.



[1]  Yinan Huang, William Lu, Joshua Robinson, Yu Yang, Muhan Zhang, Stefanie Jegelka, and Pan Li. On the stability of expressive
positional encodings for graph neural networks. arXiv preprint arXiv:2310.02579, 2023.

[2] Derek Lim, Joshua Robinson, Lingxiao Zhao, Tess Smidt, Suvrit Sra, Haggai Maron, and Stefanie Jegelka. Sign and basis invariant networks for spectral graph representation learning. arXiv preprint arXiv:2202.13013, 2022.

[3] Bohang Zhang, Lingxiao Zhao, and Haggai Maron. On the expressive power of spectral invariant graph neural networks. arXiv preprint arXiv:2406.04336, 2024.

**Questions:**

The paper states that "taking inner products can potentially lose much of the rich structural and positional information carried by vanilla Laplacian eigenvectors." However, it seems to me that the inner product, which serves as a projection matrix onto the corresponding eigenspace, should preserve all the information about that space. I thought that the added expressivity comes from using more expressive point cloud architectures rather than the inner product itself.  Could the authors provide a more detailed explanation or proof of what specific information, if any, is lost when taking inner products of eigenvectors compared to using the raw eigenvectors? Additionally could the authors clarify whether the increased expressivity comes primarily from the expressive power of point cloud architectures rather than avoiding inner products?

---

> ### Author Response · Authors · 2024-12-01
>
> Thank you for your valuable comments and questions. We respond to all the weaknesses and questions below. As we have revised our paper and given some responses in our public comments titled **"Paper revision"** (shortened as **PR**) and **"Further comments regarding Q1 of Reviewer wrYh and Q1 of Reviewer 9aT4"** (shortened as **FC**), in the following we will directly quote them if our intended response is already there.
>
> **For W1.** First, we explain why our method stands as a novel approach to incorporate Laplacian eigenvectors into MPNNs while ensuring sign-and-basis-invariance. As we have pointed out in **FC**, almost all existing methods to process Laplacian eigenvectors in a sign-and-basis-invariant fashion try to encode the pairwise inner products between Laplacian eigenvectors, which constitute an $n\\times n$ matrix for each eigenspace. It is currently unknown how to universally encode those $n\\times n$ matrices. Therefore, such methods fail to achieve universal representation of graphs. In contrast, our method provides a practical possibility to achieve universal representation of graphs, by directly encoding the Laplacian eigenvectors themselves with sufficiently powerful point cloud networks. This methodology shift marks a major novelty of our paper.
>
> Second, we explain how our approach to ensuring stability differs from the one adopted in [1]. The work [1] uses $Z(G)=\\rho\\big(\\mathbf{V}\\mathrm{diag}(\\phi\_1(\\mathbf{\\Lambda}))\\mathbf{V}^T,\\ldots, \\mathbf{V}\\mathrm{diag}(\\phi\_m(\\mathbf{\\Lambda}))\\mathbf{V}^T\\big)$ to produce a provably stable positional encoding, while we use $Z(G)=\\mathrm{MultisetEncoder}\\left(\\left\\{\\!\\!\\left\\{\\left(\\lambda\_k,f(\\mathbf{V}\_k^\\text{smooth})\\right)\\right\\}\\!\\!\\right\\}\_{k=1}^K\\right)$. One may easily see the following differences between our approach and the one in [1]: (a) we direct encode the Laplacian eigenvectors (with smoothing factors), instead of encoding the inner products between eigenvector pairs; (b) we explicitly treat different Laplacian eigenspaces separately, by first generating an encoding for each of them, and then pooling them into a multiset, while [1] only treats different Laplacian eigenspaces differently by an eigenvalue modulation factor $\\phi(\\mathbf{\\Lambda})$; (c) by properly selecting the smoothing function $\\rho$, we can control the degree of overlap between $\\mathbf{V}\_1^\\text{smooth},\\ldots, \\mathbf{V}\_K^\\text{smooth}$, which reflects the degree of parameter sharing across different eigenspaces. The approach in [1] is less flexible than ours.
>
> **For W2.** We provide below a theoretical argument for the ability of our method to capture global properties of graphs. Notice that PSRD coordinates [2] can provably capture shortest-path distances (Theorem 5.1 of [2]), i.e., they are provably aware of global properties. But PSRD coordinates are just a special case of OGE-Aug, with the specific choice of the point cloud network similar to SchNet [3] or LorentzNet [4]. Therefore, OGE-Aug has the potential to capture global properties such as shortest-path distances. Actually, we remark that any graph property that can be expressed or approximated as a polynomial of the Laplacian $\\mathbf{L}$ is computable given full spectral information of $\\mathbf{L}$.
>
> Currently, we did not conduct synthetic experiments to verify the above argument, i.e., the ability of our method to capture shortest-path distances or other long-range properties. We apologize for that, and will report our updated result in our future paper revisions or comments. On the other hand, we may still witness evidence of the good global performance of OGE-Aug from its results on several real-world benchmarks, such as the targets $U\_0$, $U$, $H$ and $G$ of QM9, as well as PCQM-Contact of the Long Range Graph Benchmark.
>
> [1] Yinan Huang, William Lu, Joshua Robinson, Yu Yang, Muhan Zhang, Stefanie Jegelka, and Pan Li. On the stability of expressive positional encodings for graph neural networks. arXiv:2310.02579.
>
> [2] Xiyuan Wang, Pan Li, Muhan Zhang. Graph as Point Set. arXiv:2405.02795.
>
> [3] Kristof T. Schütt, Pieter-Jan Kindermans, Huziel E. Sauceda, Stefan Chmiela, Alexandre Tkatchenko, Klaus-Robert Müller. SchNet: A continuous-filter convolutional neural network for modeling quantum interactions. arXiv:1706.08566.
>
> [4] Gong, S., Meng, Q., Zhang, J. et al. An efficient Lorentz equivariant graph neural network for jet tagging. J. High Energ. Phys. 2022, 30 (2022).

---

> ### Author Response · Authors · 2024-12-01
> **Official Comment by Authors (continued)**
>
> **For W3.** Actually, if we require the $f$ in OGE-Aug (Definition 4.1) to be an $O(n)$-invariant universal representation, then OGE-Aug is no less expressive than Vanilla OGE-Aug, and is thus universal. However, this makes the method too costly. Thus, we do not use an $O(n)$-invariant universal representation for $f$ in practice. We provide the details of our practically used point cloud network (and the corresponding OGE-Aug architecture) in Algorithm 1 (Appendix D.2.1), in which only tensors up to 2nd order are used. Although we cannot quantitatively determine the expressive power of the resulting model, we argue that it is no less powerful than Point Set Transformer proposed in [1], as the point cloud network we use is no less powerful than the one adopted in [1].
>
> Regarding the misalignment between our definition for stability (Definition 4.2) and Proposition 4.3, we have slightly revised our definition for stability (please refer to the 2nd bullet point of **PR** and our revised paper) to make our Proposition 4.3 fully align with it.
>
> **For W4.** We have detailed such comparison in **FC**, in which the EPNN framework of [2] is discussed. Similar to existing works such as BasisNet [3] or SPE [4], EPNN is also among the methods that try to encode inner products between Laplacian eigenvectors, instead of directly encoding the eigenvectors themselves. Further, the expressive power of EPNN is bounded by certain subgraph GNNs (and thus by 3-WL), while our method has the potential to achieve universal representation of graphs.
>
> **For W5.** Please refer to the 5th bullet point of **PR** as well as our revised paper.
>
> **For Q1.** Please refer to the 3rd bullet point of **PR**, as well as **FC**.
>
> [1] Xiyuan Wang, Pan Li, Muhan Zhang. Graph as Point Set. arXiv:2405.02795.
>
> [2] Bohang Zhang, Lingxiao Zhao, and Haggai Maron. On the expressive power of spectral invariant graph neural networks. arXiv:2406.04336.
>
> [3] Lim, Derek, et al. Sign and basis invariant networks for spectral graph representation learning. arXiv:2202.13013.
>
> [4] Huang, Yinan, et al. On the stability of expressive positional encodings for graph neural networks. arXiv:2310.02579.

---

### Official Review · Reviewer_7pZy · 2024-11-03

**Soundness:** 2
**Presentation:** 3
**Contribution:** 3
**Rating:** 6
**Confidence:** 4

**Summary:**

This paper studies the improvement of positional encodings (PEs) based on Laplacian eigenvectors, commonly used as feature augmentations to enhance the expressivity and effectiveness of GNNs.
The paper points out that existing PEs, which deal with symmetries of Laplacian eigenvectors with naive $O(p)$-group invariant encoders, may lose the full expressivity in the eigenvectors. Moreover, the paper argues that the instability resulting from the properties of eigenvectors is ignored in most research. Thus, the paper proposes a novel encoder to generate expressive and stable positional embeddings by utilizing invariant point cloud networks and incorporating smooth function for "soft splitting". Theoretical analysis and extensive experiments demonstrate the effectiveness of the proposed positional encoder.

**Strengths:**

1. The paper is well-organized and easy to follow.
2. The experimental results seem promising.
3. Authors utilize a novel strategy to softly split the eigenvectors from different eigenspaces, solving the instability constraint of most existing work.

**Weaknesses:**

1. The novelty of the method is somewhat limited, especially the direct application of existing invariant point cloud networks over the eigenvectors without any transfer challenge.
2. The paper doesn’t provide any detail about instance models of $\rho$, $\phi$, and $\psi$ (in Equations 6 and 7) implemented in the experiments, and doesn’t discuss the Lipschitz continuity of these practical models.
3. There is no detail about how OGE-Aug deals with the sign ambiguity of eigenvectors.
4. Authors ignore some essential experiments.
    - Authors don’t verify the stability of the OGE-Aug on OOD benchmarks such as DrugOOD [1], where SPE [2] is validated on this dataset.
    - There is no ablation study on the effectiveness of the proposed “soft splitting” strategy. I think it’ll be better to conduct experiments on Vanilla OGE-Aug as a control group.
    - There is no experiment studying the sensitivity of hyperparameters. For example, the authors don’t explore how different shapes of ρ influence the effectiveness and stability of the positional encodings.
    - The paper doesn’t report the running time of different positional encoding methods.
5. As mentioned in Line 346, insisting on the universality of invariant representation function f may hurt the stability of the positional encoder, however, there is no quantitative analysis nor experimental guidance about how to trade off university and stability.

[1] Ji, Yuanfeng, et al. "Drugood: Out-of-distribution dataset curator and benchmark for ai-aided drug discovery–a focus on affinity prediction problems with noise annotations." Proceedings of the AAAI Conference on Artificial Intelligence. Vol. 37. No. 7. 2023.

[2] Huang, Yinan, et al. "On the stability of expressive positional encodings for graph neural networks." arXiv preprint arXiv:2310.02579 (2023).

**Questions:**

See weaknesses.

---

> ### Author Response · Authors · 2024-12-01
>
> Thank you for your thoughtful comments and valuable suggestions. We respond to all the weaknesses below. As we have revised our paper and given some responses in our public comments titled **"Paper revision"** (shortened as **PR**) and **"Further comments regarding Q1 of Reviewer wrYh and Q1 of Reviewer 9aT4"** (shortened as **FC**), in the following we will directly quote them if our intended response is already there.
>
> **For W1.** Although we propose to apply existing invariant point cloud networks to Laplacian eigenvectors, we point out that such application is not "without any transfer challenge".
> * First, invariant point cloud networks typically process only fixed-dimensional data (for example, 3-dimensional Euclidean coordinates), while our application requires that such networks are able to deal with coordinates with different dimensions (since Laplacian eigenspaces have different dimensions). This requirement restricts the types of point cloud networks that we may use. It is also necessary to re-implement the point cloud networks (making them able to accommodate different dimensions) to adapt them to our case. As is shown in Appendix D.2.1 in our revised paper, the invariant point cloud network used in our practically implemented OGE-Aug is different from any existing one. So actually we did not directly apply an existing point cloud network, but re-designed one.
> * Second, as we have pointed out in **FC**, almost all existing methods to process Laplacian eigenvectors in a sign-and-basis-invariant fashion try to encode the pairwise inner products between Laplacian eigenvectors, which constitute an $n\\times n$ matrix for each eigenspace. It is currently unknown how to universally encode those $n\\times n$ matrices. Therefore, such methods fail to achieve universal representation of graphs. In contrast, our method provides a practical possibility to achieve universal representation of graphs, by directly encoding the Laplacian eigenvectors themselves with sufficiently powerful point cloud networks. This methodology shift is a major novelty of our paper.
> * Finally, the fact that an arbitrary invariant point cloud network can be integrated into our framework is itself a novelty of our claimed method, as it is quite flexible.
>
> **For W2.** The explicit form of $\\rho$ is given in equation (73) of our revised paper. As for $\\phi$ and $\\psi$, we remark that our practical implementation of OGE-Aug is shown as **Algorithm 1** in Appendix D.2.1 of our revised paper, which does not fully align with equation (7). Therefore, instead of discussing the Lipschitz continuity of $\\phi$ and $\\psi$ separately, we can discuss the Lipschitz continuity of the entire Algorithm 1. Algorithm 1 includes two types of layers: MLP layers on the tensor products, as well as message passing layers. The Lipschitz continuity of those layers with respect to their arguments will be upper-bounded as long as norms of the network parameters are not large. Therefore, we claim that stability results similar to Proposition 4.3 can hold for **Algorithm 1**.
>
> **For W3.** Our framework explicitly respects sign invariance. Because by definition of $O(p)$-invariant universal representations (Definition 3.3), if $f$ is such a representation and $\\mathbf{V},\\mathbf{V}'\\in \\mathbb{R}^{n\\times p}$, with $\\mathbf{V}'$ being identical to $\\mathbf{V}$ except for some columns flipping signs, then obviously there exists orthogonal matrix $\\mathbf{Q}$ (diagonal, whose diagonal elements corresponding to those sign-flipped columns are -1, otherwise 1) that makes $\mathbf{V}=\\mathbf{V}'\\mathbf{Q}$. Thus $f(\\mathbf{V})=f(\\mathbf{V}')$.
>
> **For W4, bullet point 1.** Please refer to the 8th bullet point of **PR** and our revised paper.
>
> **For W4, bullet points 2 and 3.** Please refer to the 9th bullet point of **PR** and our revised paper.
>
> **For W4, bullet point 4.** We reported our running time in Table 6 of the revised paper.
>
> **For W5.** Although we don’t have a rigorous quantitative study on how insisting on a universal $f$ could lead to stability loss, we qualitatively argue that the more complex $f$ is, the larger $J\_f$ tends to be, making the stability bound looser. This fact somewhat implies a stability loss. A quantitative analysis of this question may be conducted by measuring the constants $J\_f$ of different point cloud networks $f$, and empirically observe the effect on $J\_f$ with increasing expressivity in $f$. But this would be out of the main scope of our work.
>
> Moreover, we point out that our main consideration to reject a universal $f$ is that it would be too computationally costly. Actually, the point clout network architecture we use can theoretically reach universality as long as we use very high-order tensors. But this makes the complexity at a scale of $O(n\\cdot n^n)$ which is unacceptable. That’s why in our Algorithm 1 we restrict the tensor order to be at most 2, making the complexity polynomial.

---

> > ### Comment · Reviewer_7pZy · 2024-12-01
> > **Response to Author Rebuttal**
> >
> > Thanks for your detailed descriptions, addressing most of my concerns. I maintain my score.

---

> > > ### Author Response · Authors · 2024-12-01
> > >
> > > Thank you for thoroughly reading our responses and giving the affirmative feedbacks.

---

### Official Review · Reviewer_wrYh · 2024-11-04

**Soundness:** 2
**Presentation:** 3
**Contribution:** 3
**Rating:** 5
**Confidence:** 2

**Summary:**

The paper presents a robust orthogonal group-equivariant neural network designed to enhance graph representation using Laplacian vectors. Experiments conducted on various graphs demonstrate the competitive performance of the proposed approach.

**Strengths:**

- The writing is well-organized and generally reader-friendly.
- The theoretical analysis is comprehensive and detailed.

**Weaknesses:**

- The datasets used in the experiments are somewhat limited. The proposed method would be more convincing if validated on a wider range of datasets, such as PATTERN, CLUSTER, or MolHIV.
- Additional illustrations, such as ablation or comparison experiments (e.g., investigating the impact of the continuous smoothing function *ρ*, or examining how the method captures global properties of graphs), would provide a more thorough and solid foundation for the study.

**Questions:**

* Could you elaborate on how taking inner products may potentially lead to a loss of the rich structural and positional information provided by vanilla Laplacian eigenvectors (line 85)?
* How does the method scale to large-scale graphs? Could you discuss its scalability?

---

> ### Author Response · Authors · 2024-11-29
>
> Thank you for your insightful comments and questions. We respond to all the weaknesses and questions below. As we have revised our paper and given some responses in our public comments titled **"Paper revision"** (shortened as **PR**) and **"Further comments regarding Q1 of Reviewer wrYh and Q1 of Reviewer 9aT4"** (shortened as **FC**), in the following we will directly quote them if our intended response is already there.
>
> **For W1.** Please refer to the 7th bullet point of **PR**, and the revised paper.
>
> **For W2.** We investigated the impact of using different $\\rho$, by altering the range width $\\delta$ on which $\\rho$ has a non-zero value. Moreover, we compared the performance of OGE-Aug on ZINC with that of Vanilla OGE-Aug to verify the significance of using the smoothing function $\\rho$. Please refer to the 9th bullet point of **PR** and the revised paper.
>
> To discuss how our method captures global properties of graphs, we may notice that PSRD coordinates [1] can provably capture shortest-path distances (Theorem 5.1 of [1]), i.e., they are provably aware of global properties. But PSRD coordinates are just a special case of OGE-Aug, with the specific choice of the point cloud network similar to SchNet [2] or LorentzNet [3]. Therefore, OGE-Aug has the potential to capture global properties such as shortest-path distances. Moreover, the good performance of OGE-Aug on some real-world benchmarks, such as the targets $U\_0$, $U$, $H$ and $G$ of QM9, as well as PCQM-Contact of the Long Range Graph Benchmark, also stands as evidence for the strong capabilities of OGE-Aug to capture global graph properties.
>
> **For Q1.** Please refer to the 3rd bullet point of **PR**, as well as **FC**.
>
> **For Q2.** In our revised paper, we show that the time complexity of our practically adopted OGE-Aug is $O((n^2\\max\_j\\mu\_j+m)\\cdot\\max\_j\\mu\_j)$, where $n$ and $m$ are the number of nodes and edges, respectively, while $\\max\_j\\mu\_j$ is the maximum dimension of Laplacian eigenspaces. Therefore, our OGE-Aug scales much better than 3-WL based methods such as 1-2-3-GNN or PPGN, which have time complexity $O(n^3)$, and comparable to subgraph GNNs, which have time complexity $O(nm)$.
>
> We also empirically measure the training time per epoch of our model. The result is shown in Table 6 of our paper. We now compare it to some baselines. We take ZINC as our benchmarking dataset. In the following table, **Time** means training time per epoch. The baseline information is quoted from [1].
>
> |  | PST |  SUN|  SSWL| PPGN| Graphormer|  SAN-GPS| **OGE-Aug (ours)**|
> |:--:|:--:|:--:|:--:|:--:|:--:|:--:|:--:|
> |**Time/s** |15 |21 |45| 20| 124|  79|28|
>
> One may see that our model runs much faster than Graphormer [4] or SAN [5], even though our model also uses the graph transformer architecture with positional encodings computed from Laplacian eigenvectors. Moreover, our model is faster than subgraph GNNs like SSWL [6]. Although the graphs in ZINC have small sizes (even too small to make dense-product based methods like SUN or PPGN inefficient), the above result still partially shows the relatively good efficiency (thus potential for scaling to large-scale graphs) of our OGE-Aug.
>
>
> [1] Xiyuan Wang, Pan Li, Muhan Zhang. Graph as Point Set. arXiv:2405.02795.
>
> [2] Kristof T. Schütt, Pieter-Jan Kindermans, Huziel E. Sauceda, Stefan Chmiela, Alexandre Tkatchenko, Klaus-Robert Müller. SchNet: A continuous-filter convolutional neural network for modeling quantum interactions. arXiv:1706.08566.
>
> [3] Gong, S., Meng, Q., Zhang, J. et al. An efficient Lorentz equivariant graph neural network for jet tagging. J. High Energ. Phys. 2022, 30 (2022).
>
> [4] Chengxuan Ying, Tianle Cai, Shengjie Luo, Shuxin Zheng, Guolin Ke, Di He, Yanming Shen, Tie-Yan Liu. Do Transformers Really Perform Bad for Graph Representation? arXiv:2106.05234.
>
> [5] Devin Kreuzer, Dominique Beaini, William L. Hamilton, Vincent Létourneau, Prudencio Tossou. Rethinking Graph Transformers with Spectral Attention. arXiv:2106.03893.
>
> [6] Bohang Zhang, Guhao Feng, Yiheng Du, Di He, Liwei Wang. A Complete Expressiveness Hierarchy for Subgraph GNNs via Subgraph Weisfeiler-Lehman Tests. arXiv:2302.07090.

---

### Author Response · Authors · 2024-11-27
**Paper revision**

We thank all the reviewers for their thoughtful feedbacks and constructive suggestions. All the comments have been scrupulously considered. We are now presenting a revised version of the paper, in which we have integrated most of the reviewers' suggestions. We apologize for our late response, and hope that our revised paper could address the common concerns of the reviewers. We will continue to respond to each reviewer separately regarding other minor issues they have raised.

Here is an outline for our revision, which summarizes the changes compared with the previous version.

- **Regarding W2 of Reviewer eRwE:** Make symbols comply with ICLR format guide across the entire paper. Especially, we use different fonts for different use cases of symbol $V$: the bold upper-case $\bm{V}$ always represents a 2-D matrix, the bold lower-case $\bm{v}$ always represents a 1-D vector, while the calligraphic $\mathcal{V}(G)$ represents exclusively the node set of graph $G$. Other matrices and vectors are also made bold.
- **Regarding W3 of Reviewer 9aT4:** Slightly revise the definition of stability, making our theoretical result fully aligned with the definition. (In the revised **Definition 4.2**)
- **Regarding Q1 of Reviewer wrYh and Q1 of Reviewer 9aT4:** Restate the source of expressivity gain of our proposed method: not merely from avoiding inner products, but by using more powerful point cloud networks. (In the revised **Section 1**) We will elaborate on this point in a subsequent response.
- **Regarding W2 of Reviewer 7pZy, W3 of Reviewer 9aT4, W1~2 of Reviewer eRwE:** Include a detailed description of our practical implementation of OGE-Aug in pseudocode, as well as a brief analysis of its computational complexity, expressivity and stability. (In the revised **Appendix D.2.1**)
- **Regarding W5 of Reviewer 9aT4:** Include **Point Set Transformer (PST)** [1], a baseline that also use Laplacian eigenvectors, on QM9. (In the revised **Table 1**)
- **Regarding W4 of Reviewer eRwE:** Include **Exphormer** [2], a state-of-the-art baseline on PCQM-Contact. (In the revised **Table 4**)
- **Regarding W1 of Reviewer wrYh and W5 of Reviewer eRwE:** Additional experiments on CLUSTER, PATTERN and ogbg-molhiv, validating the state-of-the-art performance of OGE-Aug. (In the revised **Appendix D.4**)
- **Regarding W4(1) of Reviewer 7pZy:** Additional experiments on DrugOOD, verifying the good OOD performance and stability of OGE-Aug.
- **Regarding W2 of Reviewer wrYh and W4(2)(3) of Reviewer 7pZy:** Ablation studies concerning the effect of smoothing function $\rho$ as well as the shape of $\rho$.

Additionally, **regarding W3 of Reviewer eRwE**, we have provided the code to reproduce our experimental results in the supplementary material.

[1] Xiyuan Wang, Pan Li, Muhan Zhang. Graph As Point Set. ICML 2024

[2] Hamed Shirzad, Ameya Velingker, Balaji Venkatachalam, Danica J. Sutherland, Ali Kemal Sinop. Exphormer: Sparse Transformers for Graphs. ICML 2023.

---

### Author Response · Authors · 2024-11-27
**Further comments regarding Q1 of Reviewer wrYh and Q1 of Reviewer 9aT4**

Taking inner products will not lose information, **if we can find an encoder for** $n\\times n$ **matrices universal w.r.t. simultaneous row and column permutations.** That is to say, we need an encoder $f$ on $\\mathbb{R}^{n\\times n}$, such that given $\\mathbf{A}, \\mathbf{B}\\in\\mathbb{R}^{n\\times n}$, $f(\\mathbf{A})=f(\\mathbf{B})$ if and only if there exists $\\mathbf{P}\\in S\_n$ such that $\\mathbf{A}=\\mathbf{PBP}^T$. If such an $f$ exists, then we can get a universal representation of graphs by collecting a multiset $\\{\\!\\!\\{(\\lambda\_j,f(\\mathbf{V}\_j\\mathbf{V}\_j^T))\\}\\!\\!\\}$, since $\\mathbf{L}=\\sum\_{j=1}^K\\lambda\_j\\mathbf{V}\_j\\mathbf{V}^T\_j$.

But the real problem is that **existing methods that take inner products** (for example, BasisNet [1], SPE [2], Fürer invariant [3] or EPNN [4]) **fail to find such universal encoders for $n\\times n$ matrices.** For example,

- BasisNet uses $f(\\mathbf{A})=\\text{2-IGN}(\\mathbf{A})$, which has the architecture of $f=R\\circ\\sigma\\circ L\_K\\circ\\sigma\\circ\\cdots\\circ\\sigma \\circ L\_1$. Each $L\_m$ has the form of $L\_m(\\mathbf{A})\_{ii}=\\mathrm{Linear}\\Big(A\_{ii},\\sum\_jA\_{ij},\\sum\_jA\_{ji},\\sum\_jA\_{jj},\\sum\_{j\\ne k}A\_{jk}\\Big)$ and $L\_m(\\mathbf{A})\_{ij}=\\mathrm{Linear}\\Big(A\_{ij},A\_{ji},A\_{ii},A\_{jj},\\sum\_k A\_{ik},\\sum\_k A\_{ki},\\sum\_k A\_{jk},\\sum\_k A\_{kj},\\sum\_k A\_{kk},\\sum\_{k\\ne l}A\_{kl}\\Big)$ for $i\\ne j$. The final layer $R$ is $R(\\mathbf{A})=\\mathrm{Linear}\\Big(\\sum\_i A\_{ii},\\sum\_{i\\ne j}A\_{ij}\\Big)$.
- Fürer invariant does not directly encode the multiset $\\{\\!\\!\\{(\\lambda\_k, f(\\mathbf{V}\_k\\mathbf{V}\_k^T)\\}\\!\\!\\}$. It lets $P\_{ij}^{(k)}=(\\mathbf{V}\_k\\mathbf{V}\_k^T)\_{ij}$, and $P\_{ij}=\\mathrm{concat}(P^{(1)}\_{ij},\\ldots, P^{(K)}\_{ij})$ where the concatenation is over all $P^{(k)}\_{ij}$ corresponding to different distinct eigenvalues $\\lambda\_1,\\ldots, \\lambda\_K$, and those $P^{(k)}\_{ij}$ are ordered in increasing order of $\\lambda\_k$. Then it takes the graph representation as $f(G)=f(\\mathbf{V}\_1\\mathbf{V}\_1^T,\\ldots, \\mathbf{V}\_K\\mathbf{V}\_K^T)=\\{\\!\\!\\{\\{\\!\\!\\{P\_{ij}:j\\in[n]\\}\\!\\!\\}:i\\in[n]\\}\\!\\!\\}$.
- EPNN does not directly encode the multiset $\\{\\!\\!\\{(\\lambda\_k, f(\\mathbf{V}\_k\\mathbf{V}\_k^T)\\}\\!\\!\\}$, either. Instead, it uses a message-passing-style encoder $f(G)=\\text{MessagePassing}(\\mathbf{X}, \\mathbf{P})$, where $\\mathbf{X}$ is the usual node feature, $\\mathbf{P}$ replaces the usual adjacency matrix $\\mathbf{A}$ by $P\_{ij}=\\{\\!\\!\\{(\\lambda\_k,(\\mathbf{V}\_k\\mathbf{V}\_k^T)\_{ij})\\}\\!\\!\\}\_{k=1}^K$, which somewhat looks like the Fürer invariant.

Although all above methods ensure permutation invariance and sign-basis invariance, none of the above proposes a way to **fully leverage the information in** $\\mathbf{V}\_1\\mathbf{V}\_1^T,\\ldots, \\mathbf{V}\_K\\mathbf{V}\_K^T$ (i.e., to universally encode them), even though we know that the set of $K$ $n\\times n$ matrices contains the full information of the graph. Actually, Bohang [4] has shown that EPNN is the most expressive one among those existing inner-product based methods, but EPNN is still upper-bounded by subgraph GNNs.

Based on the above discussion, we may answer **Q1 of Reviewer wrYh**: it is true that taking inner products between eigenvectors does not lose any information, but **it is true only when one uses a universal $n\\times n$ matrix encoder to encode such inner product matrices**, which is not the truth in previous works. Similarly, the answer to **Q1 of Reviewer 9aT4** is that the additional expressivity of our model comes from **using more expressive point cloud networks**, instead of avoiding inner products. Yet avoiding inner products makes it easier to build universal graph encoders: we circumvent the need of universally encoding $n\\times n$ matrices, but instead try to directly encode the $n\\times \\mu\_k$ “coordinates” within each Laplacian eigenspace, which can be easily realized by existing point cloud networks. Thus, our work actually finds a way to implement such a permutation and sign-basis invariant, yet universal graph encoder out of Laplacian eigenvectors, while existing works have failed. This is part of our main contribution.

[1] Lim, Derek, et al. "Sign and basis invariant networks for spectral graph representation learning." *arXiv:2202.13013* (2022).

[2] Huang, Yinan, et al. "On the stability of expressive positional encodings for graph neural networks." *arXiv:2310.02579* (2023).

[3] Fürer, Martin. "On the power of combinatorial and spectral invariants." *Linear algebra and its applications* 432.9 (2010): 2373-2380.

[4] Zhang, Bohang, Lingxiao Zhao, and Haggai Maron. "On the Expressive Power of Spectral Invariant Graph Neural Networks." *arXiv:2406.04336* (2024).

---

### Author Response · Authors · 2024-12-04

Dear Reviewers,

We sincerely appreciate your time and effort devoted to reviewing our paper. As the rebuttal period is soon coming to an end, this message is to gently notify you of our latest comments posted on OpenReview. We kindly request your attention to our comments, and look forward to your valuable feedbacks.

Thank you again for your invaluable contributions to the evaluation of our paper.

---

### Meta-Review · Area_Chair_qZdz · 2024-12-21

**Metareview:**

This paper addresses the problem of how to incorporate Laplacian eigenvectors into message-passing neural networks to maximize expressivity while maintaining invariance/equivariance. The authors propose two main architectures to achieve this based on positional encodings from Laplacian eigenvectors. They improve the stability by incorporating a smooth function for "soft splitting" the eigenvectors. Theoretical analysis and extensive experiments demonstrate the effectiveness of the proposed positional encoder.

The reviews for this paper were borderline. The main weaknesses are the numerical evaluation, scalability, and little novelty since the reviewers 7pZy and 9aT4 claim that there are several similar methods in the literature.

**Additional Comments On Reviewer Discussion:**

The authors give a very nice summary of the changes they implemented in response to the reviews. However, none of the reviewers raised their scores after the discussion period.

---

### Decision · Program_Chairs · 2025-01-22

Reject